# Asymmetric water transport in dense leaf cuticles and cuticle-inspired compositionally graded membranes

Aristotelis Kamtsikakis [1], Johanna Baales[2], Viktoria V. Zeisler-Diehl[2], Dimitri Vanhecke[1], Justin O. Zoppe[1], Lukas Schreiber [2✉] & Christoph Weder [1✉]

Most of the aerial organs of vascular plants are covered by a protective layer known as the cuticle, the main purpose of which is to limit transpirational water loss. Cuticles consist of an amphiphilic polyester matrix, polar polysaccharides that extend from the underlying epidermal cell wall and become less prominent towards the exterior, and hydrophobic waxes that dominate the surface. Here we report that the polarity gradient caused by this architecture renders the transport of water through astomatous olive and ivy leaf cuticles directional and that the permeation is regulated by the hydration level of the cutin-rich outer cuticular layer. We further report artificial nanocomposite membranes that are inspired by the cuticles' compositionally graded architecture and consist of hydrophilic cellulose nanocrystals and a hydrophobic polymer. The structure and composition of these cuticle-inspired membranes can easily be varied and this enables a systematic investigation of the water transport mechanism.

[1] Adolphe Merkle Institute, University of Fribourg, Fribourg, Switzerland. [2] Institute of Cellular and Molecular Botany, Department of Ecophysiology, University of Bonn, Bonn, Germany. ✉email: lukas.schreiber@uni-bonn.de; christoph.weder@unifr.ch

Directional transport is ubiquitous in natural organisms, which use this mechanism for water collection and retention[1,2]. Asymmetric transport is also of great technological relevance, in applications that range from water harvesting[3] to the separation of chemicals[4,5] to functional clothing[6]. One strategy to achieve directional transport (and other functions) is to use functionally graded materials, such as multilayered or compositionally asymmetric architectures[7–11].

A compositional gradient is also the basic design principle of the cuticles of land plants[12,13], which have the primary function of preventing desiccation[14–16]. Cuticles mainly consist of the amphiphilic long fatty acid-based polyester cutin[17] and also contain hydrophobic epi- and intra-cuticular waxes that are primarily located on the exterior side (Fig. 1a, b)[14–16]. In several plant species, polysaccharides, such as cellulose and pectin, extend from the epidermis into the cuticle[12] and are postulated to provide polar sorption sites that promote the transport of water and other polar compounds[16,18–21]. Depending on the investigated plant species, the reticulate zone formed by these polysaccharides transverses the cuticular matrix or gradually fades towards the exterior[12,13], leading, along with the waxes, to a transversal polarity gradient, as demonstrated by bilateral desorption studies[22].

Membrane theory predicts that such compositionally graded architectures should render the water transport characteristics of cuticles asymmetric, with preferential diffusion from the polysaccharide-rich interior towards the waxy outer surface[9,11]. However, since the main function of cuticles is water conservation[16], this directionality would be surprising. Indeed, isolated measurements of insect[23] and astomatous ivy (Hedera helix) leaf cuticles[24] suggest that the water permeability in the inward direction may be higher. In view of the previously established compositionally graded architecture of astomatous olive (Olea europaea) leaf cuticles[25,26] and their thoroughly investigated chemical composition[25–29], we hypothesized that these membranes should display a particularly pronounced asymmetric water permeability. Interestingly, previous transport studies on olive cuticles[27,30] and also artificial cuticle-inspired multilayer membranes[31–35] have not explored this aspect. Thus, we embarked to investigate the directionality of the water transport through olive leave cuticles and nanocomposites inspired by their graded structure (Fig. 1a–c and Supplementary Fig. 1). Comparative studies of the water transport through ivy cuticles were also conducted. We find that water transport is indeed asymmetric in all systems and further demonstrate that the permeation is regulated by the hydration status of the membranes. While the directionality in the artificial membranes follows the polarity gradient as expected, we find an opposite preference in the olive cuticles. Our data show that in the biological membranes, the water transport is governed by the hydration of the cutin-rich exterior side and not the polysaccharide-rich interior side, which is ecologically meaningful.

## Results and discussion
**Compositionally graded cuticle-inspired membranes**. We originally expected (but, as discussed below, eventually disproved) that the directional water transport behavior of olive cuticles is mainly caused by the lipophilic waxes that are preferentially located towards the outer cuticular side, and the hydrophilic polysaccharides at the interior of the cuticles (Fig. 1b). Thus, we approximated the complex architecture of the cuticle by a compositionally graded two-component nanocomposite. A hydrophobic polymer matrix was used in lieu of the non-polar waxes and lipophilic portions of cutin, hydrophilic cellulose nanoparticles assume the function of the cuticular polysaccharides,

and the polar portions of the cutin matrix were omitted (Fig. 1c). On account of its hydrophobic nature and excellent film-forming properties, poly(styrene)-block-poly(butadiene)-block-poly(styrene) (SBS) was used as the matrix. SBS is an amorphous (Supplementary Fig. 2e), physically cross-linked rubbery block copolymer, which phase segregates into glassy poly(styrene) and soft poly(butadiene) domains. Somewhat akin to the natural cuticles, in which the impermeable domains formed by the waxes render the water transport tortuous, the glassy poly(styrene) domains of the SBS are relatively impermeable to water, whereas the more mobile poly(butadiene) domains allow the diffusion of small molecules[36]. Rod-like cellulose nanocrystals (CNCs)[37] isolated from cotton, with average dimensions of $128 \times 14$ nm (Supplementary Fig. 2a–d), were employed as water-transporting filler, mimicking the polar polysaccharides found in cuticles. The crystalline nature of CNCs (Supplementary Fig. 2e) suppresses transport through the nanoparticles[38], but their incorporation into hydrophobic polymers was previously shown to cause an increase of the water sorption and in some cases also transport along their hydrophilic surface[39–41]. We prepared SBS/CNC nanocomposite membranes with a 5–15 wt% content of CNCs and a thickness between ca. 30 and 120 μm via solvent casting from tetrahydrofuran. The limited colloidal stability of the CNCs in this solvent and their gravitational sedimentation during drying[42,43] (Supplementary Note 6 and Supplementary Fig. 3a–c) were exploited to create a CNC concentration gradient in the transversal direction.

In order to monitor the transversal CNC distribution, we acquired attenuated total reflection infrared (ATR-IR) spectra of both sides of the membranes (~1.7 μm penetration depth) and of the reference materials (neat SBS film, CNC powder) (Fig. 1d, Supplementary Fig. 4 and Supplementary Table 1). The spectra recorded on the top side of all SBS/CNC membranes are void of the characteristic cellulose signals, notably the −OH stretch between 3600 and 3100 cm$^{-1}$, the C–O–C stretch at 1162 cm$^{-1}$, the glucose ring stretch at 1110 cm$^{-1}$, the sharp stretching vibrations of C–O/C–C at 1054 and 1031 cm$^{-1}$, and the −OH bending vibrations at 663 cm$^{-1}$ (Supplementary Table 1), while these signals are clearly present in all spectra recorded on the bottom sides of the membranes. To complement the ATR-IR experiments, we also mapped the CNC distribution by Raman microscopy of the membranes' cross-sections and expressed the relative CNC concentration as the ratio of the intensities of the signals associated with the CNCs' C–O stretch (1096 cm$^{-1}$) and the SBS' CH$_2$ deformation (1437 cm$^{-1}$) (Fig. 1e and Supplementary Fig. 5b–d). Figure 1e, which shows data for an ~33-μm-thick membrane of a composite containing 10 wt% CNCs, reveals that the relative CNC concentration is highest at the bottom side and drops continually towards the top. Similar results were obtained for all compositions (Supplementary Fig. 5c, d) and thicknesses (Supplementary Fig. 6b). The data seem to suggest that the gradient steepens with the CNC content and membrane thickness, consistent with the accelerated CNC sedimentation expected for the related process conditions, i.e., higher CNC concentration and increased evaporation times when casting thicker membranes under the same casting conditions (Supplementary Fig. 6d, e). We also utilized scanning electron microscopy (SEM) to image the membranes' cross-sections. The SEM images reveal a uniform and smooth morphology for the neat SBS reference membrane (Fig. 1f), whereas graded textures are present in the SBS/CNC membranes (Fig. 1g and Supplementary Fig. 7), with a rough morphology that is characteristic of CNC composites towards the bottom side[44], and a smooth texture that suggests the absence of CNCs towards the top. Thus, a simple casting protocol afforded membranes with a steep concentration gradient of CNCs along the transversal direction.

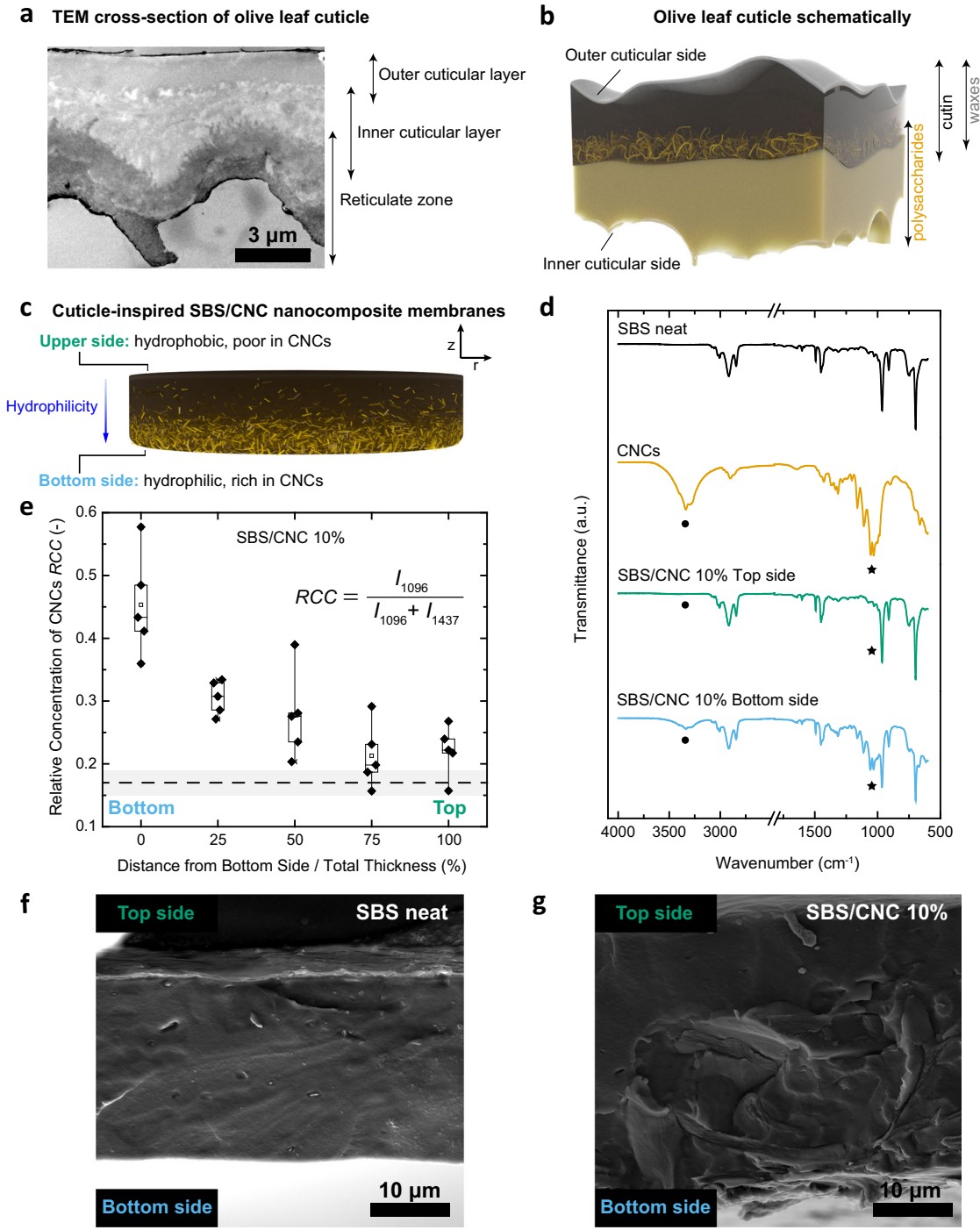

**Fig. 1 Transversal composition gradients in olive cuticles and cuticle-inspired artificial membranes. a** Cross-section of an olive cuticle imaged with TEM. **b**, **c** Schematics (not to scale) of the graded architecture in the **b** olive cuticle and **c** SBS/CNC membranes. **d** ATR-IR spectra of SBS, CNCs, and an SBS/ CNC composite at the top and bottom side of the membrane. Signals marked with dot and star correspond to the CNCs' –OH and C–O/C–C vibrations. **e** Box plot showing the relative concentration of CNCs (RCC) as a function of transversal position ($n = 5$ in each position) in a SBS/CNC membrane determined by Raman microscopy. The RCC was calculated from the intensities ($I$) of the CNCs' C–O (1096 cm$^{-1}$) and SBS' CH$_2$ (1437 cm$^{-1}$) vibration bands. Whiskers extend to min and max values, box edges show 25–75 percentiles, center line represents median, and hollow square represents mean. The dashed line and gray band are the mean ± s.d. ($n = 3$ in each position) of the RCC of the SBS reference. **f**, **g** SEM images of the cross-section of **f** SBS and **g** SBS/CNC membranes. All composite membranes shown here were ~33 μm and contained 10 wt% CNCs.

**Asymmetric water transport through SBS/CNC membranes.** The water transport characteristics of the SBS/CNC membranes were investigated as a function of direction, CNC content, membrane thickness, and relative humidity at the donor side (RH$_D$) using gravimetric dry (for RH$_D$ = 75 and 85%) and wet cup (for RH$_D$ = 100%) methods (Fig. 2a, b and Supplementary Fig. 8a, b).

Initial experiments were carried out on membranes having a thickness of 33 ± 5 μm and with a relative humidity on the receiver side (RH$_R$) of 0%. When the CNC-rich bottom sides are exposed to the donor (Fig. 2c), the water permeability increases with the CNC content and RH$_D$, suggesting that the CNCs promote, as intended, water transport through the hydrophobic SBS (Fig. 2f,

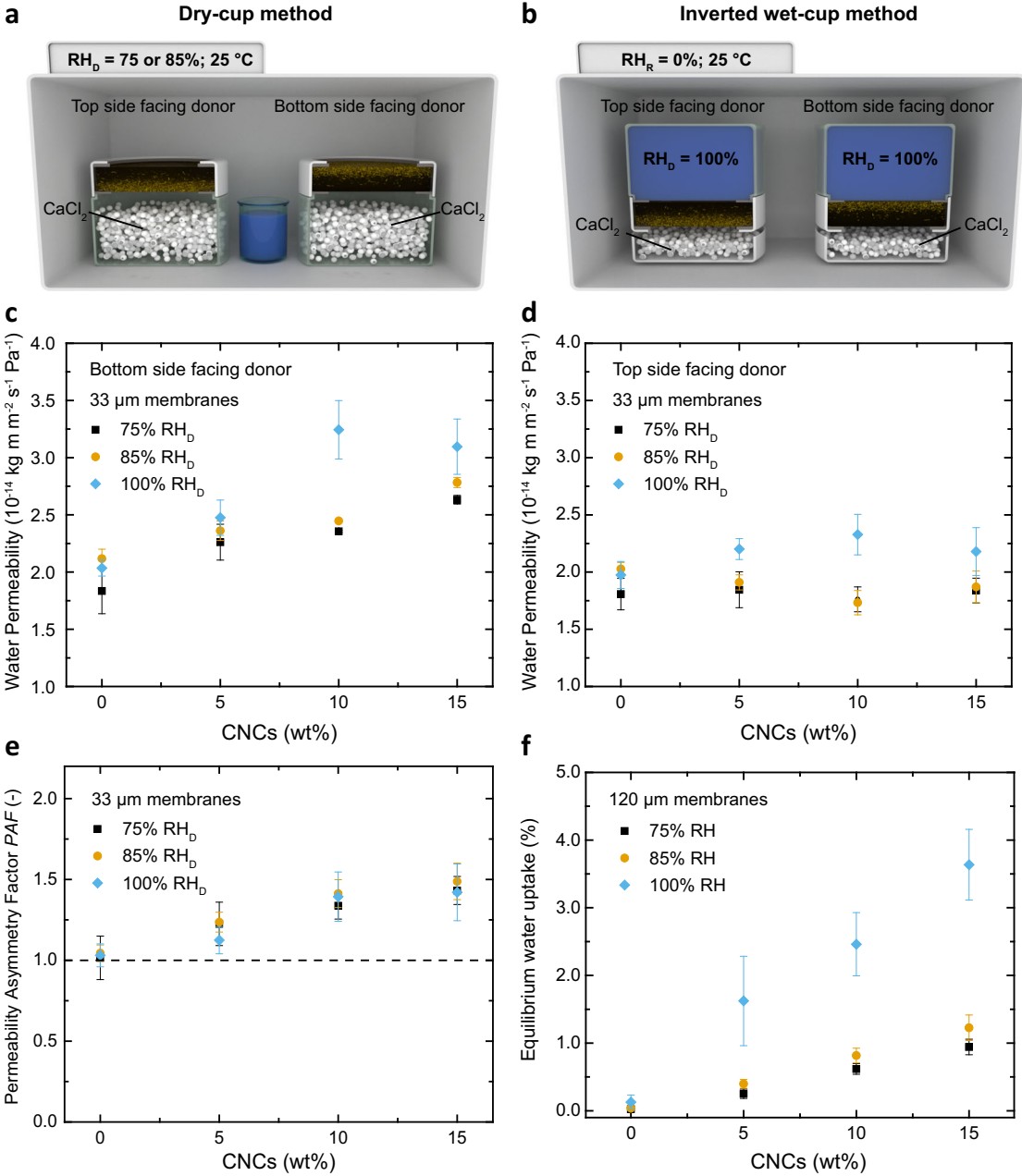

**Fig. 2 Asymmetric water permeability of compositionally graded SBS/CNC membranes. a, b** Schematic (not to scale) representation of the gravimetric **a** dry cup and **b** inverted wet cup method that were used to determine the water permeability from both sides of the SBS/CNC nanocomposite membranes at 75, 85, and 100% relative humidity in the donor ($RH_D$) at 25 °C. **c, d** Plots showing the water permeability of SBS and SBS/CNC membranes (thickness 33 ± 5 μm) as a function of the relative humidity in the donor ($RH_D$) and the CNC content with **c** the bottom (i.e., rich in CNCs) and **d** the top side exposed towards the donor, respectively. **e** Plot of the permeability asymmetry factor (PAF) established from the data shown in **c** and **d**. **f** Plot of the equilibrium water uptake of ~120-μm-thick SBS and SBS/CNC membranes as a function of CNC content and relative humidity. All experiments were conducted at 25 °C. Reported values are the mean ± s.d. of $n = 3$ measurements from different samples.

Supplementary Notes 8–10 and Supplementary Fig. 8c, d). For example, for $RH_D = 100\%$, the permeability rose from 2.0 ± 0.1 (neat SBS) to $3.1 ± 0.2 × 10^{-14}$ kg m m$^{-2}$ s$^{-1}$ Pa$^{-1}$ (15 wt% composite). The apparent water diffusion coefficient $D$ decreases with increasing CNC content, due to the tortuosity imparted by the filler (Supplementary Note 9 and Supplementary Fig. 8c). When the membranes are placed with the CNC-free top side towards the donor (Fig. 2d), the permeability values are similar for all compositions and $RH_D$ values, reflecting that the SBS-dominated top layer limits the water uptake and permeation. The asymmetric water transport characteristics of the SBS/CNC membranes can be

expressed by a permeability asymmetry factor (PAF), defined as the ratio of the permeabilities measured in the two directions. As shown in Fig. 2e, the PAF of membranes with the same thickness (~33 μm) increases with the CNC content from 1.03 (neat SBS) to 1.42 (15 wt% CNCs), independent of the relative humidity at the donor ($RH_D$). As an intrinsic property, the water permeability of (homogeneous) membranes should not be influenced by their geometry[45], but a dependence on the thickness has been reported for dense membranes—both hydrophobic, but primarily hydrophilic—and this was explained with swelling effects[46–48]. Accordingly, the water permeability (measured with the wet cup method)

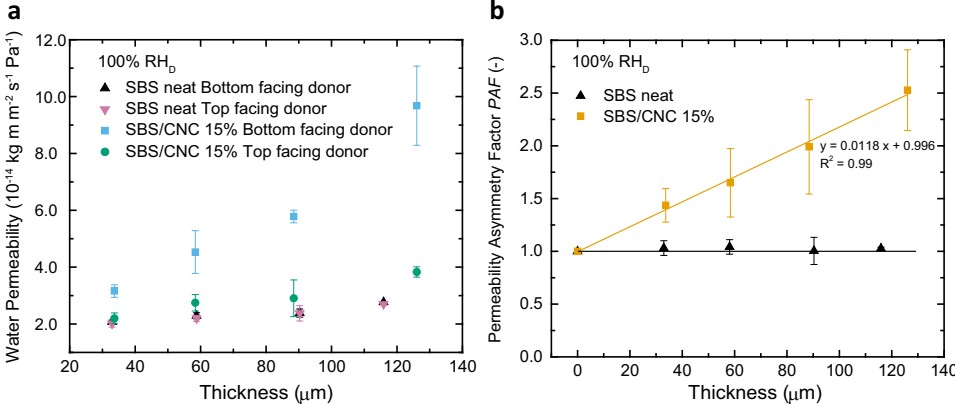

**Fig. 3 Asymmetric water permeability of compositionally graded SBS/CNC membranes.** Plots of **a** the impact of membrane thickness on water permeability and **b** the corresponding permeability asymmetry factor (PAF) of membranes made from neat SBS and SBS/CNC 15 wt% using the wet cup gravimetric method ($RH_D = 100\%$) (data from **a**). All experiments were conducted at 25 °C. Reported values are the mean ± s.d. of $n = 3$ measurements from different samples.

of the neat SBS membrane only slightly increases with increasing thickness from ~33 to ~120 μm (Fig. 3a) and the transport remains symmetric, as reflected by the PAF ($1.00 \pm 0.13$) (Fig. 3b). By contrast, the permeability of the SBS/CNC membranes containing 15 wt% CNCs increased by 206% (bottom side facing the donor) and 74% (top side facing the donor) when the thickness was increased from 33 to 120 μm (Fig. 3a), while the asymmetry factor increases linearly with the thickness from 1.44 to 2.53 (Fig. 3b). This behavior is consistent with a higher concentration of sorption-dictating CNCs towards the bottom side of thicker membranes, which as discussed above is the result of the increase in evaporation times when casting thicker samples (Supplementary Fig. 6d, e).

**Asymmetric water transport through olive and ivy cuticles.** The cuticular membranes studied here were enzymatically isolated from the adaxial (upper) side of *O. europaea* (olive) leaves, which are free of stomatal pores and allow the investigation of purely diffusional transport through dense matter. The cuticles feature a thickness of ca. 10 μm, except around the occasionally occurring trichomes, which locally increase the thickness to ca. 30 μm (Supplementary Fig. 13 and Supplementary Note 2). The previously reported graded structure of olive cuticles[25,26] was confirmed by transmission electron microscopy (TEM), focused ion beam-SEM (FIB-SEM), and ATR-IR spectroscopy (Supplementary Fig. 1 and Supplementary Note 13). Moreover, TEM and FIB-SEM images show that the cuticular layer underneath the trichomes is continuous, suggesting that these sites do not represent "leaky" pores (Supplementary Fig. 13 and Supplementary Note 12). The possibility to remove the waxes by solvent extraction allowed us to investigate their influence on the water transport by measuring cuticles in their native (i.e., with waxes) and wax-free state. In view of their fragility and small size, we conducted radiolabeled water permeation experiments with $^3H_2O$ as a tracer molecule[49,50] and determined the $^3H_2O$ permeance for $RH_D = 100\%$ and $RH_R = 2$ or 100% (Fig. 4a, b)[50], i.e., conditions with a large and without a humidity gradient. Experiments were conducted both in the physiological configuration found in nature, i.e., with the inner cuticular side of the membranes towards the donor (Fig. 4a) (outward transport) and the inverse arrangement, i.e., with the outer cuticular side facing the donor (inward transport) (Fig. 4b), to probe any asymmetry (Fig. 4c). Because the permeance is an area-normalized quantity, a precise determination of the sample's surface area is important. Indeed, theoretical and experimental studies have shown that roughness

increases the effective surface area and thereby the permeances[51,52]. While the outer cuticular side is relatively smooth and its surface area can accurately be calculated from the lateral dimensions, the cutinized residual cell wall that protrudes from the inner cuticular side (also known as internal cuticular pegs[12,53]) increases the effective surface area of olive cuticles, and thereby the sorption and permeance, by ca. 54% relative to a flat surface (Supplementary Figs. 1g, h and 10, and calculations in Supplementary Note 12). This effect has been previously demonstrated in bilateral desorption studies with several plant cuticles[22]. The higher surface roughness of the inner cuticular side leads to an overestimation of the permeance when this side faces the donor and this skews a direct comparison of the water transport from the two sides. To compensate for this effect, we calculated the permeances and the asymmetry factors (PAFs) for a flat and a rough inner cuticular side whose effective surface area was estimated as outlined above (Fig. 4a, c, Supplementary Table 3 and Supplementary Table 4, unless mentioned, values quoted in the text are corrected).

When the inner cuticular side faces the donor ($RH_D = 100\%$) and is fully hydrated (outward transport), the cuticles with waxes (i.e., native membranes) exhibit a water permeance of $2.01 \pm 1.04$ and $4.04 \pm 1.40 \times 10^{-10}$ m s$^{-1}$ for dry ($RH_R = 2\%$) and wet ($RH_R = 100\%$) receiver side conditions (Fig. 4a and Supplementary Table 3), i.e., the $^3H_2O$ permeance doubles when the outer cuticular side is hydrated from the receiver side. Consistent with earlier findings[20], the $^3H_2O$ permeance increases by a factor of 20 when waxes are extracted from the native cuticles (Fig. 4a, b and Supplementary Tables 3 and 4) highlighting the well-known protective role of waxes as a water transport barrier[16]. Also, in this case, a twofold-increase of the permeation is observed when the outer cuticular side is hydrated from the receiver side ($RH_R = 100\%$). The data collected when the membranes were flipped over, i.e., with the outer cuticular side facing the donor (inward transport), paint a similar picture, except for the notable difference that the permeance of the wax-free cuticles is the same for $RH_R = 2$ and 100% (Fig. 4b). This result reflects again that hydration of the cutin-dominated outer cuticular side—in this case from the donor side when $RH_D = 100\%$—governs the water permeation and not, as originally expected, the hydration of the polysaccharide-rich inner cuticular side. As a consequence, transport through the cuticles is essentially symmetric for both wax-containing and wax-free cuticles when $RH_R = RH_D = 100\%$ and the membranes are fully hydrated, whereas asymmetric

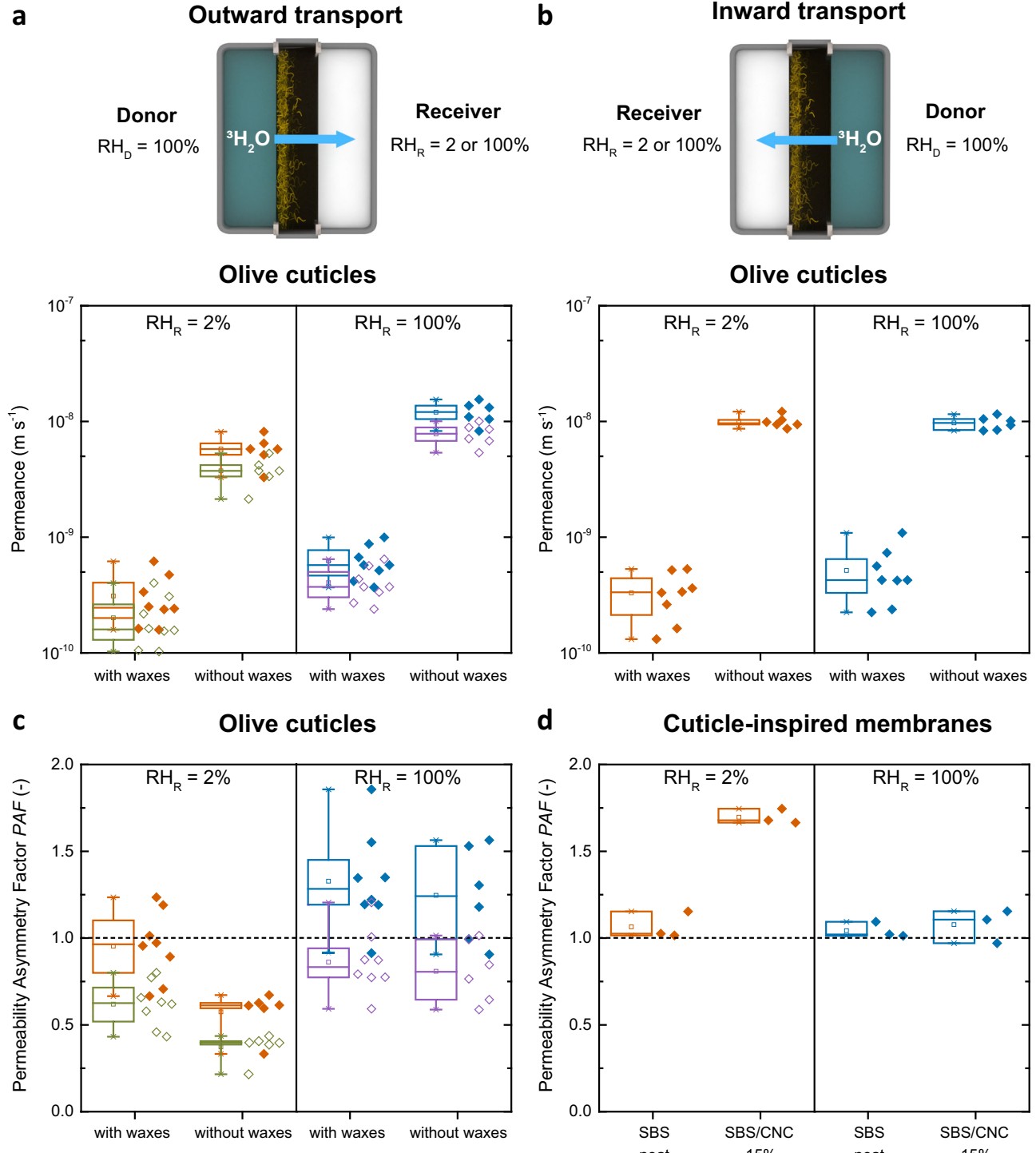

**Fig. 4 Water transport through olive cuticles and cuticle-inspired membranes measured by radiolabeled $^3H_2O$ permeation tests. a, b** Schematics (not to scale) of the measurement configurations and box plots showing the outward and inward transport of $^3H_2O$ through olive cuticles with and without waxes under varying relative humidity conditions at the receiver ($RH_R = 2$ or 100%) and membrane orientations. The geometry was varied so that **a** the inner cuticular side (outward transport) and **b** the outer cuticular side (inward transport) faced the donor ($RH_D = 100\%$). Permeances were calculated assuming the inner cuticular side as flat (solid orange and blue symbols and boxes) or rough (open green and purple symbols and boxes). **c, d** Box plots of the permeability asymmetry factor PAF of **c** ~20 μm olive cuticles and of **d** ~33-μm-thick membranes of SBS or SBS/CNC containing 15 wt% CNCs determined by $^3H_2O$ permeation experiments with the CNC-rich bottom side and the top side facing the donor ($RH_D = 100\%$). In all box charts, whiskers extend to min and max values, box edges show 25–75 percentiles, center line represents median, and hollow square represents mean. The experiments were conducted at 25 °C with $n = 8$ (olive with waxes), $n = 6$ (olive without waxes), and $n = 3$ (SBS neat, SBS/CNC 15%) different membranes.

transport is observed in case of a dry receiver ($RH_R = 2\%$) (Fig. 4c and Supplementary Tables 3 and 4). In this case, the PAFs of wax-containing ($PAF = 0.62 \pm 0.13$) and wax-free ($PAF = 0.37 \pm 0.08$) membranes are both smaller than unity, i.e., water transport is enhanced when the outer cuticular side faces the donor and swells. While the absolute value of the PAF varies considerably on the inner cuticular side surface area employed to calculate the permeance, the wax-free cuticles (Fig. 4c) display such a pronounced directionality that irrespective of the data analysis, the preferred water transport direction is unequivocally from the outside to the inside.

Noting that ivy (*H. helix*) leaf cuticles are also astomatous, trichome-free, and compositionally graded[12] (Supplementary Fig. 11), but do not exhibit pronounced inner cuticular side pegs[12,53] that complicate the data analysis (Supplementary Fig. 10 and Supplementary Note 12), we also conducted radiolabeled $^3H_2O$ water permeation tests through cuticles that had been isolated enzymatically from the adaxial side of ivy leaves (Supplementary Fig. 12). Also in this case, the inward water permeance is higher than the outward transport ($PAF = 0.60 \pm 0.16$; $RH_R = 2\%$) (Supplementary Fig. 12a, b and Supplementary Table 6) and, similarly to the behavior observed for the olive cuticles, the outward permeance of $^3H_2O$ grows by ca. 33% upon increasing $RH_R$ from 2 to 100% (Supplementary Fig. 12a and Supplementary Table 6). Thus, while our transport measurements were conducted under different conditions than the ones used by Schieferstein and Loomis[24], we observed a similar directionality of water transport ($PAF < 1$, i.e., inward transport > outward transport). Moreover, our findings that the outward water transport in olive and ivy cuticles is humidity-dependent is consistent with previous works using other species, e.g., citrus (*Citrus aurantium* L.)[49], eggplant (*Solanum melongena*)[49], beech (*F. sylvatica* L.)[54], and others[50,55].

To explore if the water transport through the artificial membranes is also humidity-dependent, we conducted radiolabeled water permeation experiments for $33 \pm 5$-µm-thick membranes of SBS and SBS/CNC with a CNC content of 15 wt% (Fig. 4d and Supplementary Table 5). The $^3H_2O$ permeance through the neat SBS membrane ($1.03$–$1.22 \times 10^{-9}$ m s$^{-1}$) is hardly affected by the receiver conditions and remained symmetric ($PAF = 1.04 \pm 0.07$ and $1.06 \pm 0.08$ for $RH_R = 2$ and 100%), consistent with its symmetric structure and low water uptake (Fig. 4d and Supplementary Table 5). The nanocomposite membrane displays a permeability asymmetry factor ($PAF = 1.69 \pm 0.37$ for $RH_R = 2\%$) (Fig. 4d), which is comparable to the value determined with the gravimetric wet cup method ($PAF = 1.44 \pm 0.16$, Fig. 2c, f). Also in this case the transport becomes symmetric when both sides of the membranes are hydrated ($PAF = 1.06 \pm 0.30$ for $RH_D$ and $RH_R = 100\%$, Fig. 4d). A comparison shows that the permeances ($3.04$–$3.46 \times 10^{-9}$ m s$^{-1}$) are similar when the CNC-rich side is hydrated (i.e., facing a wet donor or receiver) and the only configuration in which a lower permeance is observed ($1.80 \times 10^{-9}$ m s$^{-1}$) is the one when the CNC-rich side faces the dry receiver and the membrane is prevented from swelling with water (Supplementary Table 5). Thus, the fundamental principle (but not the mechanism, see below) leading to asymmetric permeation and humidity-dependent transport behavior in olive cuticles and SBS/CNC nanocomposite membranes is the same. In both cases the transport is directional when the membranes are dry and one of their sides shows strong humidity-dependent permeation properties as expected from polymer membrane theory[9]. This asymmetry vanishes when the membranes are fully hydrated in both the biological and artificial systems.

**Mechanical properties and plasticization of olive cuticles.** The different transport behavior of the biological and artificial membranes investigated here appears to be linked to a crucial role that the cutin matrix plays in the cuticles, and which is absent in the simplified SBS/CNC system. In the latter, the SBS matrix hardly takes up water (Fig. 2f), and therefore exhibits a permeability coefficient that is humidity-independent (i.e., ideal behavior) (Fig. 2c, d). As a result, the CNCs drive, as intended, the water transport in the artificial membranes. By contrast, it is well known that water plasticizes cutin, leading to a reduction of the glass transition temperature and related changes of the mechanical properties[53,56–59]. The resulting increase of the polymer chain mobility and free volume should lead to an increase of the permeability coefficient, as reported for many synthetic polymers[60], including multi-component structures[9], in which this effect is at play. A hydration-related increase of the permeance has also been observed for other cuticles[49,50,54,55] and was explained with the formation of "aqueous pores" that facilitate water transport[20,49,50], which may be related to cutin plasticization. In order to investigate this further, we probed the viscoelastic properties of olive and ivy cuticles using dynamic mechanical analysis (DMA) (Fig. 5a–c). The DMA traces of the wax-containing olive cuticles show that the storage modulus $E'$ drops in two steps as samples are heated from −100 to 150 °C (Fig. 5a), reflecting two broad phase transitions between ca. −35 and 35 °C, as well as 35 and 150 °C (Fig. 5b), which we interpret as glass transition ($T_g$) of the cutin matrix and melting of aliphatic waxes, respectively (for representative DMA traces of ivy cuticles see Supplementary Fig. 12c). The damping factor tanδ traces put the maxima of the olive cuticle transitions at −7 and 102 °C, respectively (Fig. 5b and Supplementary Table 7). The corresponding traces of the wax-free olive cuticles show only one prominent broad glass transition centered around 24 °C (perhaps with a shoulder at −8 °C) (Fig. 5b) and the storage modulus reduction is much more pronounced (Fig. 5a and Supplementary Table 7), which reflects the notable mechanical reinforcement imparted by the waxes in the native cuticles. This effect has also been observed for other cuticles with similar wax composition[27,29,61]. It is at first glance surprising that the DMA trace of the wax-free cuticles shows a transition that is not observable in the wax-containing samples (Fig. 5b). However, the two tan δ peaks of the wax-containing cuticles are not well separated, and it is well possible that the phase transition observed for the wax-free membranes is also present, but hidden. Alternatively, it is possible that the waxes plasticize or anti-plasticize the cutin and that their removal affects the $T_g$.

DMA measurements also allow a comparison of the storage moduli $E'$ of cuticles that had been equilibrated under ambient conditions (50% RH) and were subsequently wetted with water in situ (Fig. 5c). Gratifyingly, the storage modulus $E'$ value of both, wax-containing and wax-free olive cuticles, measured at 25 °C is substantially reduced upon wetting. In the case of native wax-containing cuticles the storage modulus $E'$ drops from $309 \pm 51$ to $182 \pm 43$ MPa and in the case of the wax-free membranes from $134 \pm 26$ to $21 \pm 4$ MPa (Fig. 5c). The latter value is comparable to the storage modulus $E'$ recorded at elevated temperatures (e.g., $32 \pm 13$ MPa at 130 °C), indicating that swelling of the cuticles with water changes the physical properties in a manner that is similar to raising the temperature above $T_g$. Thus, aqueous swelling has the same effect as reducing $T_g$ to below ambient temperature[62]. The plasticizing effect is also observed in the wax-containing cuticles, but although the total water uptake is similar for both wax-containing ($6.1 \pm 2.4$ %) and wax-free ($6.9 \pm 1.3\%$) membranes (both measured at 97% RH, n = 3), the mechanical

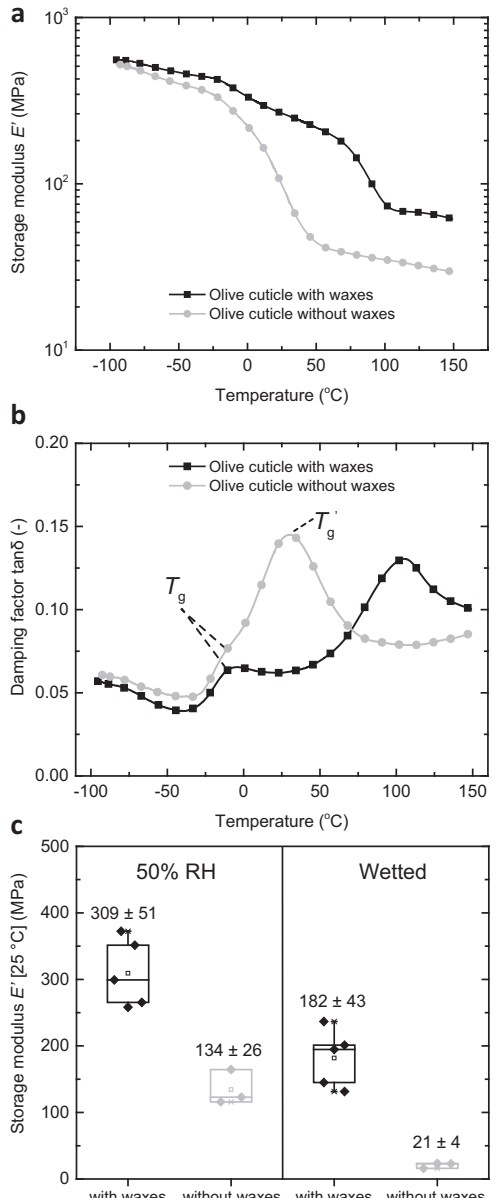

**Fig. 5 Dynamic mechanical analysis (DMA) data of olive leaf cuticles with and without waxes. a, b** Representative DMA traces of **a** the storage modulus $E'$ and **b** the damping factor tan $\delta$ displaying the cuticular phase transitions. **c** Box chart displaying the storage modulus $E'$ of olive cuticles at 25 °C conditioned at 50% RH and wetted with water showing the plasticizing effect of water. In all boxes, the whiskers extend to min and max values, box edges show 25–75 percentiles, center line represents median, and hollow square represents mean. Reported values are the mean ± s.d. of paired measurements at ambient and wet conditions ($n = 5$ olive with waxes; $n = 3$ olive without waxes). Note that the data were analyzed using the overall average thickness that is "inflated" by the protruding trichomes; as a result, the reported storage modulus $E'$ values are systematically underestimated by a factor of ca. 2–3 (see also "DMA measurements" section).

contrast is less pronounced (Fig. 5c). This is consistent with the fact that aqueous swelling does not reduce the stiffness imparted by the waxes, which in turn are melted upon heating at elevated temperatures (Fig. 5a). The same plasticizing effect was observed for the wax-containing ivy cuticles (Supplementary Fig. 12). While the DMA data clearly reflect that water plasticizes the

cuticles as a whole (Fig. 5c), the data do not permit us to draw an unequivocal conclusion how the various components (polysaccharides, cutin, and other non-extractable components) contribute to this effect. However, taken together with the permeation data (Fig. 4a, b), which show that the directionality in olive cuticles is dictated by the hydration status of the outer, cutin-rich side (Fig. 4a–c), and findings by others, who have reported the plasticization of cutin upon hydration,[53,56–59] the DMA data support the hypothesis that the plasticization of cutin is presumably a major driver of the observed effects.

In summary, we have shown that the water permeation properties of enzymatically isolated plant cuticles and artificial membranes inspired by their graded structure can be asymmetric and have demonstrated that this propensity is driven by selective swelling. Intriguingly, in the biological membranes, the water-responsive, plasticizable cutin-rich outer cuticular side appears to control the water transport directionality, which may be ecologically meaningful. In dry conditions, the water permeability is low and this helps the plant to retain water. During fog and rain, cuticles swell from the outer side and are plasticized, changing the mechanical and transport characteristics of the membrane. This mechanism may help plants to dissipate excess water to the environment or help them to take up moisture through the cuticles if the internal water potential is low, as speculated by others[20,63]. The absolute permeability and asymmetry factor of the artificial membranes reported here are of similar magnitude as the values observed for the wax-free biological membranes. Because the limited contrast of mass-transport rates in the artificial system studied here is related to the moderate swelling, it appears that it can be readily and substantially increased by substituting the crystalline CNC filler against an amorphous, hydrophilic, water-plasticizable glassy polymer, such as for example poly(vinyl alcohol) nanofibers[64,65] or other less crystalline polysaccharide-based fibers. Importantly, our comparative study has highlighted the significance of the water-switchable nature of the cuticular membranes. It should be feasible to incorporate the functionality that this feature imparts into future artificial membranes, which in addition to the elements utilized here contain a water-plasticizable component. Membranes that exhibit the directional, switchable water transport feature reported here should be useful for smart packaging and other applications that require directional mass-transport properties such as in membrane reactors, fuel cells, and drug delivery systems.

## Methods

**Materials**. Poly(styrene)-*block*-poly(butadiene)-*block*-poly(styrene) (SBS with 30 wt% styrene, weight-average molecular weight, $M_w$ ~140,000 g mol$^{-1}$ by size exclusion chromatography; density, $\delta = 0.94$ g cm$^{-3}$) was purchased from Sigma-Aldrich (now Merck). Tetrahydrofuran (THF) (99.5%), ethanol (95%), calcium chloride (CaCl$_2$) (anhydrous, granular ~1–2 mm), sodium chloride (NaCl), potassium chloride (KCl), potassium sulfate (K$_2$SO$_4$), and sulfuric acid (H$_2$SO$_4$) (95–98%) were supplied from Merck and used without further purifications. In-house deionized (DI) water was utilized in all cases unless otherwise stated. CNCs were isolated by sulfuric acid hydrolysis from Whatman® No. 1 filter paper (Sigma-Aldrich) by following previously reported procedures[66]. Briefly, Whatman® No. 1 filter paper (30 g) was cut into rectangular pieces (ca. 1 × 1 cm) and hydrolyzed with sulfuric acid (64 wt%, 430 g of diluted acid) for 45 min at 45 °C under vigorous mechanical stirring with a glass impeller. After the hydrolysis, the diluted acid solution was quenched with ca. 500 g of cold DI water and the CNCs were isolated from the acid by three successive centrifugations (15 min each cycle, ca. 30 000g), before they were redispersed in 600 mL of DI water and further purified by dialysis (Spectra/Por 4 dialysis tubes MWCO 12–14 kDa) against DI water for 7 days. After the dialysis, the aqueous suspension of CNCs was filtered under vacuum using a glass filter pore size 1 to remove large aggregates followed by a brief sonication step (2 min, 10% amplitude, Branson Sonifier SFX550). The isolated aqueous suspension of CNCs was freeze-dried with a Telstar LyoQuest lyophilizer (−50 °C, 0.2 mbar) for 10 days. The dimensions of CNCs were determined to be 128 ± 55 nm × 14 ± 4 nm × 9 ± 3 (length × width × aspect ratio) by transmission electron microscopy images (Supplementary Fig. 2a–d and Supplementary Note 1), their

crystallinity index (CI) was calculated to be 82% using the Segal method (Supplementary Fig. 2e and Supplementary Note 4) and the amount of sulfate half-ester groups was 254 mmol kg$^{-1}$ by conductometric titration (Supplementary Fig. 2f and Supplementary Note 5).

**Enzymatic isolation and treatment of plant cuticles**. Olive cuticles were enzymatically isolated from the adaxial side of full-grown leaves harvested in an olive grove near Siena (Italy). Ivy cuticles were isolated from full-grown leaves collected outside the Institute building in Bonn (Germany). Enzymes (cellulase and pectinase dissolved in 10$^{-2}$ M citric buffer adjusted to pH 3.0 at 2 vol% concentrations) were obtained from Erbslöh (Germany). The isolated cuticles were washed in DI water and in borate buffer (10$^{-2}$ M, pH 9.0). Finally, they were carefully flattened under a gentle air stream and stored at room temperature in Petri dishes. For wax extraction, isolated cuticles were immersed for 24 h in chloroform:methanol (50:50 v:v). For transpiration measurements, tritiated water ($^3H_2O$; specific activity: 37 MBq g$^{-1}$; Hartmann Analytik, Braunschweig) was used. Amounts of $^3H_2O$ diffused across the cuticle were mixed with scintillation cocktail (Ultima Gold XR, PerkinElmer) and measured by scintillation counting (LSA Tri-Carb 2800TR, PerkinElmer) with a 2-sigma error of 2%. For FIB-SEM imaging of the membranes, they were stained using solutions of 2 wt% OsO$_4$ (Merck), 1.5 wt% K$_4$[Fe(CN)$_6$] (Sigma-Aldrich, now Merck) in DI water and 1 wt% thiocarbohydrazide (Merck) in DI water.

**Preparation of SBS/CNC nanocomposites by solvent casting-evaporation**. A solution of SBS (1.6 wt%) in THF and freeze-dried CNCs (0–15 wt% of final nanocomposite) were combined in a sealed round-bottom flask, the mixture was stirred for 30 min at room temperature, and the flask was transferred to a sonication bath (Sonoswiss SW3H ultrasonic bath) equipped with a water cooling system. After 2 h of sonication, the mixtures were immediately casted into preheated poly(tetrafluoro ethylene) (PTFE) Petri dishes with a diameter of 80 mm and the solvent was evaporated at 50 °C in a ventilated oven overnight. The membranes were then cooled to room temperature and peeled off with a spatula. Their mean thickness, measured at random spots ($n = 10$) with a digital micrometer (IP 65, Mitutoyo), ranged between 30 and 130 μm, depending on the selected mass of the casting components (e.g., ~300 and ~800 mg total mass of polymer and CNCs for 30- and 120-μm-thick membranes). The membranes were stored in a closed container at room conditions.

**Verification of transversal heterogeneity by ATR-IR spectroscopy**. Fourier transform IR spectra of the artificial and cuticular membranes were recorded with a PerkinElmer Spectrum 65 spectrometer in ATR mode (Universal ATR model, ~1.7 μm depth of penetration). The SBS/CNC nanocomposite membranes and the cuticles were deposited on the ATR crystal, fixed with a mechanical clamp and spectra were recorded on both sides of the membranes to detect the compositional variation of the two sides (artificial membranes: Supplementary Fig. 4 and Supplementary Fig. 6a; olive cuticles: Supplementary Fig. 1a, b). The acquired spectra were an average of eight accumulations at a 4 cm$^{-1}$ resolution in a wavenumber range between 4000 and 600 cm$^{-1}$. Collection and analysis of the IR spectra were conducted with the infrared spectroscopy software Spectrum 10 (PerkinElmer). To determine the evolution of the relative CNC content at the bottom side of SBS/CNC membranes with 15 wt% CNCs as a function of thickness, we used the same instrumentation and acquisition parameters. The ratio of baseline-corrected absorbance intensities of CNCs' –OH (3338 cm$^{-1}$) (Supplementary Table 1) and SBS CH = CH (3005 cm$^{-1}$)[67] was used to show the increasing relative CNC content at the bottom side of the membranes as a function of thickness (Supplementary Fig. 6d, e).

**Verification of transversal heterogeneity by Raman microscopy**. Raman spectroscopy was utilized to corroborate the transversal heterogeneity of the membranes and reveal the distribution profile of the CNCs in the cross-section of the SBS/CNC membranes. The latter were imbibed in ethanol-filled syringes, immersed in liquid nitrogen, and subsequently cryo-fractured in liquid nitrogen using a pincer. Cross-sections of the cryo-fractured membranes were secured between glass slides and examined under a HORIBA Jobin Yvon LabRAM HR800 Raman microprobe spectrometer (polarized 633 nm excitation laser, 800 mm focal length; spectral window 400–4000 cm$^{-1}$; spectral resolution 0.3 cm$^{-1}$; ×50 objective). The number of accumulations per spectrum was set to 5 and their duration to 5 s at a maximum laser intensity (20 mW). Linear mapping of the CNCs in the membranes was conducted by acquiring spectra initially at the bottom and the top side of the samples and then by focusing the laser beam at the cross-section at three intermediate positions (25, 50, and 75% of the total thickness; bottom side $x = 0$%; top side $x = 100$%). Five replications ($n = 5$) per spot ($5 \times 5 = 25$ total unique spots) were acquired to investigate the variation between different spots of the same distance from the bottom side of the membranes. The LabSpec 6 software (HORIBA Scientific) was utilized to obtain the Raman spectra, smooth and baseline correct the raw data, and generate Raman images. The relative concentration of CNCs (RCC) in the membranes for each spot was calculated

according to Eq. (1):

$$RCC = \frac{I_{1096}}{I_{1096} + I_{1437}} \tag{1}$$

where $I_{1096}$ and $I_{1437}$ corresponded to the baseline-corrected absolute signals of CNCs (C–O stretch)[68,69] and SBS (CH$_2$ deformation)[70], respectively. Box charts of the calculated RCC as a function of the relative distance from the bottom side were constructed using Origin 2016G (OriginLab). In all box charts, whiskers extend to min and max values, box edges show 25–75 percentiles, center line represents median, and hollow square represents mean.

**Water permeability measurements (ASTM E96)**. Standard test methods for water vapor transmission of materials according to ASTM E96 (dry and inverted wet cup methods) were followed to measure the water permeability of SBS and SBS/CNC membranes[71]. For details on the experimental procedure, see Supplementary Note 2.

**Permeability measurements with radiolabeled water**. Radiolabeled water ($^3H_2O$) permeation experiments were conducted using a previously established method[50]. Artificial SBS and SBS/CNC 15 wt% membranes (ca. 30 μm) or isolated cuticular membranes with and without waxes were mounted between two stainless steel adapter rings with central openings of 0.28 cm$^2$. This allowed to fix the transpiration chambers filled with 800 μL $^3H_2O$ (on average 10$^{13}$ dpm m$^{-3}$) subsequently to both sides of the adapter rings and thus ensured that combined replicates could be measured at different receiver humidities (RH$_R$ = 2 or 100%) and with different membrane orientations (inner or outer cuticular side facing the donor). The integrity of the membranes was checked by using an "ethanol test" as previously described[72]. Briefly, a small drop of ethanol, when added to the outer cuticle surface, spreads spontaneously and penetrates the membrane through macroscopically invisible cracks and pores, if present in the cuticle. Consequently, membranes with cracks turn immediately dark, since the refraction of light is changed. Only membranes which pass the ethanol test were used in transpiration experiments. The interfaces between the membranes, the adapter rings, and the transpiration chambers were sealed with PTFE-paste (Roth). The adapter rings, closed on one side with transpiration chambers containing radiolabeled water (donor), were turned upside down and placed with the other side of the adapter ring on scintillation vials (receiver). Different receiver humidities in each vial were adjusted by adding either 250 μL water (100% RH$_R$) or glycerol (2% RH$_R$) to the bottom of the scintillation vials. These solvents simultaneously served as sinks for the diffused radiolabeled water across the membranes. Transpiration was measured at a constant temperature of 25 °C. Before starting transpiration measurements, the membranes were incubated on scintillation vials with either water or glycerol for at least 24 h to achieve humidity equilibration between the membranes and the gas phase. Transpiration kinetics at the two different receiver humidities (RH$_R$ = 2 or 100%) and with the two different orientations (inner or outer cuticular side facing the donor, i.e., outward and inward transport, respectively) of the membranes were measured for a total time of 8 h investigating combined replicates. At each sampling time (0, 2, 4, 6, and 8 h) scintillation vials were replaced by new ones. A scintillation cocktail was added to the sampled scintillation vials and the quantity of permeated $^3H_2O$ was measured by scintillation counting. For each transpiration kinetic, the amount of $^3H_2O$ which had diffused across the membrane was plotted as a function of time (coefficients of determination of the regression lines ($R^2$) were always better than 0.98) to determine the flux $J_{^3H_2O}$. The $^3H_2O$ permeance was calculated using Eq. (2):

$$\mathcal{P}_{^3H_2O} = \frac{J_{^3H_2O}}{A(C_D - C_R)} \tag{2}$$

where $A$ was the available surface area of the inner or outer cuticular side when the respective sides were facing the donor (see the main text and Supplementary Note 12). $C_D$ and $C_R$ are the concentrations of $^3H_2O$ in the donor and receiver, respectively. Since the concentration of $^3H_2O$ in the receiver $C_R$ is negligible compared to the donor $C_D$ on the time scale of the measurements, only $C_D$ was used for the calculations as previously reported[50]. $^3H_2O$ permeance is reported in m s$^{-1}$. To quantify the directional transport through the cuticular membranes, we used the PAF defined as the ratio of transpiration in the two membrane orientations (Eq. 3):

$$PAF_{cuticle} = \frac{P_{outward}}{P_{inward}} \tag{3}$$

where outward and inward denote the respective mass-transport directions through the membranes. $^3H_2O$ permeation experiments were also conducted with artificial SBS/CNC membranes and the asymmetry factor was defined as the transport ratio with the bottom and top side facing the donor, respectively (Supplementary Information Eq. 4).

**DMA measurements**. DMA measurements were conducted on a dynamic mechanical analyzer (TA Instruments, Model Q800) with a film-tension clamp in tensile mode with constant oscillation parameters (1 Hz frequency; 15 μm amplitude). For more details on the experimental procedure, see Supplementary Note 3.

**Statistics and reproducibility**. To verify the cuticular transversal architecture with TEM (representative cross-section of olive cuticles displayed in Fig. 1a and Supplementary Fig. 1c, d; ivy cuticular cross-section in Supplementary Fig. 11b), and to investigate the inner and outer side morphology of the cuticular membranes (representative SEM images displayed in Supplementary Figs. 1e–h and 10b–e), we used four different samples (2 × with waxes, 2 × without waxes) of each species. For the FIB-SEM images (Supplementary Figs. 1e, f and 13), we used two wax-containing olive membranes. For the size analysis of the CNCs (Supplementary Fig. 2a–d), we used five different images of the same sample and a total of 120 CNCs were used to determine the CNC dimensions. For the supplementary POM (Supplementary Fig. 3d–g) and SEM images (Supplementary Figs. 6c and Fig. 7) of the SBS/CNC composites, we used two membranes of each composition. Tests for statistical significance were conducted using Microsoft Excel 2016 (see also source data file).

**Reporting summary**. Further information on research design is available in the Nature Research Reporting Summary linked to this article.

## Data availability

The datasets are available through the Zenodo Sharing platform [https://doi.org/10.5281/zenodo.4461817]. Source data are provided with this paper.

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

## Acknowledgements

This project has received funding from the European Union's Horizon 2020 research and innovation program under the Marie Skłodowska-Curie grant agreement No. 722842 and the Adolphe Merkle Foundation. J.O.Z. would like to thank the Swiss National Science Foundation Ambizione Grant No. PZ00P2_167900. L.S. acknowledges funding by the DFG (German Research Foundation). The authors also thank Dr. Pierre Brodard for providing access to the Raman microscope, Dr. Miguel Spuch Calvar for creating the schematics, Beat Haenni of the Microscopy Imaging Center (University of Bern) for the cuticular microtomes, and Dr. Bodo Wilts and Prof. Michael Mayer for their valuable comments on the manuscript.

## Author contributions

A.K., J.O.Z., L.S., and C.W. conceived and designed the research. A.K., J.B., and V.V.Z.-D. conducted the experiments and analyzed the data. A.K. prepared and characterized the artificial membranes. J.B. and V.V.Z.-D. isolated the plant cuticles and performed the water permeation experiments with radiolabeled water. D.V. and A.K. carried out the electron microscopy. A.K., D.V., L.S., J.O.Z., and C.W. interpreted the results. All authors contributed to writing the article, and read and approved the final manuscript.

## Competing interests

The authors declare no competing interests.
