## [Peer Review File · Nature Communications]

Reviewers' comments:

Reviewer #1 (Remarks to the Author):

The authors of the manuscript "asymmetric water transport in olive cuticles and cuticle-inspired, compositional graded membranes" have compiled a convincing data set in respect of the generation membranes having a polarity gradient (in the following abbreviated PG-membranes) and explaining the water transport across such membranes that will have very useful applications. These membranes were called plant cuticle mimics because the authors were inspired by plant cuticles. The authors studied the water transport through plant cuticles in comparison, focusing on olive leaf cuticles, revealing that olive leaf cuticles did not have the same behavior as the PG-membranes, but when the outside was fully hydrated water molecules moved in the opposite direction. This could be seen in the presence and absence of apolar waxes. The authors discuss the outcome by the swelling behavior of cutin based on the literature, but no further studies have been performed to explain the difference between the cuticle and the PG-membranes. The studies on the PG-membrane nevertheless brings a great advance to our understanding of directional transport of water molecules across cuticular membranes. However, the cuticular membrane has not been successfully mimicked.

The authors should have realized during the studies that the initial statement that the cutin matrix is non-polar is – to say the least - a strong oversimplification since the cutin polymer is composed of a high percentage of oxygenated fatty acids, thus, has amphipolar components. Therefore, repetition of this statement in the view of the presented data set is rather irritating. The direct comparison between cuticles and PG-membranes leaves the reader rather puzzled about the different outcome of the studies. Nevertheless, the data set PG-membranes brings very useful aspects to understanding better asymmetric membranes, including cuticles. Therefore, the similarities and differences should be better worked out though the entire manuscript and the PG-membranes should not be called cuticle mimics.

It is difficult for a reader who is not in the cuticle field to understand the schematic diagram of the cuticle in comparison to the structure of the olive cuticle next to it. The olive cuticle has a particular complex multilayered structure. The olive cuticle structure would need much more explanation, i.e. annotation in the picture, in the main figure and extended data set. In addition, it is puzzling that the authors show different structures of the olive leaf cuticle without mentioning the frequency with which they were observed as well as conditions or developmental stage, etc. when they appear in the extended data set.

For this reason, the olive leaf cuticle may not be the best suitable cuticle in comparison to the two-component system of the PG-membrane. Why did the authors not chose a simpler cuticle, such as the tomato fruit cuticle, having a uniform structure somewhat comparable to schematic diagram (except that the wax is not crystalline) and of which the composition/structure is also much better understood. Studying the complex olive leaf cuticle raises then the question whether the complex ultrastructure of which the chemical-structural basis is not known may be, at least in part, the source of the particular water transport behavior. Therefore, it would be useful to also include data of a simple-structured cuticle to show whether it behaves as the olive leaf cuticle under identical settings.

A more detailed discussion with additional schematic diagrams explaining the cuticle as a two/multi component system of a polar and an amphipolar polymer both having (different?) swelling properties in contrast to the PG-membrane of which the apolar part does not have swelling properties would be useful. Of course, best would be to generate an artificial membrane having such properties and add water transport data of it. But this may be for future research.

Minor comments:

Figure 1b; water drop would need explanation in legend or remove droplet. It is irritating as the cutin matrix of the olive leaf cuticle has also a round structure on the surface, that has nothing to do with the water drop.

Figure 1f and 1g as well as in extended data: Orange arrows are not explained in legend.

Furthermore, explain the role of waxes in barrier properties more precisely. Maybe wax influences also the swelling properties of the cutin polymer?

Reviewer #2 (Remarks to the Author):

Kamtsikakis and coworkers analyze the asymmetric diffusion of water across composite layers such as the olive leaf cuticle and bio-inspired SBS/CNC membranes. While the manuscript address an interesting and important question, I am concerned that some of the conclusions for the cuticle layer are drawn with limited structural analysis and experimental support. I have the following specific comments:

1) The main text describes rather well the experiments made to ascertain the structure of the SBS/CNC membranes. In contrast, the text and supplementary material are much weaker when it comes to the structural description of the cuticle. I believe, the following issues will have to be addressed:

(a) I am unable to infer what I am seeing in Fig. 1a. To me, it looks as if a wax/cutin layer (above) has detached for the outer wall of an epidermal cell (below). Also, the authors have not labeled the layers that are visible and attempted to draw a parallel with the diagram of Fig. 1b. In the supplementary material, the authors provide more information on the structure of the cuticle but again it is very difficult to know what was, in concrete terms, the structure of the cuticle used in their experiments.

(b) Extended data Fig. 1 makes reference to two types cuticles: Type 1 (lamellate/reticulate) and Type 2-3 (faintly lamellate, amorphous and gradually reticulate architecture). Do the author know what was the dominant type for their cuticle samples? What are the relative areas occupied by these cuticle types in the tested samples? There is a striking difference in the thickness of the outer lamellate layer between Type 1 and Type 2-3. Does this structural difference, or any other structural differences between the types, have a functional effect on the diffusion of water?

(c) There are no stated attempts to characterize the integrity of the isolated cuticles. Since the cuticle has low conductivity to water, even small, hard-to-notice, defects could easily govern the diffusion of water. Along the same line, how does the thickness of the different layers vary spatially? Again, reduction in thickness of the water impermeable layers, even over small areas, could easily control the overall behavior of the cuticle sample.

(d) The chemical characterization of the cuticle is minimal and none of it is presented in the figures of the main text. I believe additional characterization would be useful. For example, I would have liked to see staining for specific polysaccharides (cellulose, pectins) and waxes. According to the authors, the cuticle is, on average, 20µm thick. Therefore, it should be possible to resolve the distribution of some chemical components both in the plane of the cuticle AND across its thickness.

(e) Finally, why are the authors studying the cuticle of *Olea* specifically? The authors do not state evidence that the cuticle of this species has some special properties. Do they have reasons to believe that *Olea* has a cuticle better adapted for asymmetry diffusion of water? It seems strange to talk about bio-inspired membranes from *Olea* if the species is not unique in some way.

2) The authors did not mention that the adaxial leaf surface of *Olea* has trichomes. Some of them are seen in Ext. Data Fig. 1g,h. What was the density of these trichomes? Can the authors confirm that these trichomes do not function as "water pores" in isolated cuticles? Trichomes of xerophytes have been shown to transport water asymmetrically, favoring the inward flow of water as observed in their work. See for example,

Biebl, R. (1964). Zum Wasserhaushalt von *Tillandsia recurvata* L. und *Tillandsia usneoides* L. auf Puerto Rico. *Protoplasma*, 58(3), 345-368.

Benzing, D. H., & Burt, K. M. (1970). Foliar permeability among twenty species of the Bromeliaceae. *Bulletin of the Torrey Botanical Club*, 269-279.

Barthlott W, Capesius I. 1974 Wasserabsorption durch blatt- und sproßorgane einiger Xerophyten. *Z. Pflanzenphysiol.* 72, 443-455.

Rundel, P. W. (1982). Water uptake by organs other than roots. In *Physiological plant ecology II* (pp. 111-134). Springer, Berlin, Heidelberg.

Wang, X., Xiao, H., Cheng, Y., & Ren, J. (2016). Leaf epidermal water-absorbing scales and their absorption of unsaturated atmospheric water in *Reaumuria soongorica*, a desert plant from the northwest arid region of China. *Journal of Arid Environments*, 128, 17-29.

Raux et al. (2020). Design of a unidirectional water valve in *Tillandsia*. *Nature communications*, 11(1), 1-7.

The last paper reach very similar conclusions as this manuscript, including the effect of humidity on the asymmetry in the permeability (PAF). Can the authors explain why this was not considered as an explanation for their observations?

3) The authors use the results of Fig. 3 to claim that diffusion across the cuticle is asymmetric when the relative humidity is low (2%) on the receiver side. However, this conclusion is reached only when the data are corrected for a postulated effect of excess area on the ICS. Looking at the RAW data measured from the UNTREATED cuticle (i.e. the closed, uncorrected, symbols in Fig. 3a,b), one sees that the measured permeance is not affected by the orientation of the cuticle at RH = 2% and only slightly so for RH = 100%. A statistical analysis of the effect of membrane orientation on the permeance would almost certainly show that the RAW distributions are not statistically different. Unfortunately, such analyses were not done. Instead, the authors cite other studies that have claimed an effect of surface roughness on the permeance. Based on these studies, the initial distributions are rescaled when the ICS is on the donor side. It is only after this correction is made that the authors are able to make the claim mentioned above. The conclusion reached by the authors hinged 100% on their hypothesis that the raw data must be rescaled to take into account the ICS roughness. Although it is quite possible that surface roughness, in some context, can affect permeance, I believe the authors are obligated to show that surface roughness is important for the intact cuticle of *Olea*. Without this experimental support, the authors have effectively taken two distributions that are not statistically different and made them different without much more than a hunch that it ought to be so.

Based on these three comments, I believe the central claim that "transport of water through stomatous olive leaf cuticles is indeed directional and that the permeation is regulated by the hydration level of the cutin-rich layer" is not properly established.

Some minor comments:

1) line 26: "Directional transport is ubiquitous in natural organisms, which use this mechanism for water collection and retention" This statement is supported by citing ref(11), yet the cited paper is about the directional migration of droplets on spider silk, which is very different from the meaning given to "directional transport" in the remainder of the paper. Also, to my knowledge, spiders do not use this mechanism "for water collection and retention". Please cite a more relevant paper.

2) line 222: "Our data suggest that the water losses of plants through the cuticles double in wet conditions, which is ecologically meaningful." I believe this statement is incorrect. The permeability to water, not the water losses, is doubled under wet conditions. Presumably, the effect on water losses is negligible since the environment is wet and therefore the gradient for water diffusion is weak.

3) The authors rely heavily on acronyms. I believe the text would be easier to follow if fewer acronyms were used.

4) It is good that the authors are using a standard protocol to measure the water permeability of the membranes. However, the authors refer to "wet cup" and "dry cup" set-ups in the main text without explaining clearly what they are. Perhaps it would be useful to add the diagrams of Extended Data Fig. 8 a,b to the figures in the main text so that a reader unfamiliar with this type of experiments could understand what was done. Also, I could not find curves of cup weight vs time which were used to compute the permeance. It would be good to show some of these curves so that the reader can verify the linearity of the gravimetric method.

5) line 186: "While our data show that the PAF depends considerably on the ICS surface area employed to calculate the permeance, the wax-free (MX) cuticles (Fig. 3c) display such a pronounced directionality that irrespective of the data analysis, the preferred water transport direction is unequivocally from the outside to the inside." I believe the first part of this statement is misleading. The authors have not demonstrated with their data that surface roughness is relevant for permeance or PAF measurements. For their data to SHOW dependence, they would have had to perform experiments where surface roughness is varied and its effect on permeance measured. The authors have not done such experiments. Instead, they assume that surface roughness might be important (based on publications on other systems) and analyze their dataset in two different ways. Please clarify the first part of this statement to avoid confusion.

Response to Reviews and Reference to Changes (NCOMMS-20-18275-T)

Reviewer #1

The authors of the manuscript “asymmetric water transport in olive cuticles and cuticle-inspired, compositional graded membranes” have compiled a convincing data set in respect of the generation membranes having a polarity gradient (in the following abbreviated PG-membranes) and explaining the water transport across such membranes that will have very useful applications. These membranes were called plant cuticle mimics because the authors were inspired by plant cuticles. The authors studied the water transport through plant cuticles in comparison, focusing on olive leaf cuticles, revealing that olive leaf cuticles did not have the same behavior as the PG-membranes, but when the outside was fully hydrated water molecules moved in the opposite direction. This could be seen in the presence and absence of apolar waxes. The authors discuss the outcome by the swelling behavior of cutin based on the literature, but no further studies have been performed to explain the difference between the cuticle and the PG-membranes. The studies on the PG-membrane nevertheless brings a great advance to our understanding of directional transport of water molecules across cuticular membranes.

We thank the referee for their efforts to assess our manuscript, the concise summary, and the positive evaluation of our work, especially the assessment that the data set for the artificial membranes is convincing, that such membranes may be useful, and that their study advances the understanding of water transport across cuticular membranes.

However, the cuticular membrane has not been successfully mimicked.

Indeed, the artificial membranes are not strictly cuticle-mimics, but rather membranes that were *inspired* by the transversal polarity gradient present in many plant cuticles including olives. We have edited the manuscript throughout to reflect this (e.g.: Abstract page 1, page 3, page 12).

The authors should have realized during the studies that the initial statement that the cutin matrix is non-polar is – to say the least - a strong oversimplification since the cutin polymer is composed of a high percentage of oxygenated fatty acids, thus, has amphipolar components. Therefore, repetition of this statement in the view of the presented data set is rather irritating. The direct comparison between cuticles and PG-membranes leaves the reader rather puzzled about the different outcome of the studies. Nevertheless, the data set PG-membranes brings very useful aspects to understanding better asymmetric membranes, including cuticles. Therefore, the similarities and differences should be better worked out though the entire manuscript and the PG-membranes should not be called cuticle mimics.

The referee raises very valid points. We did realize that the cutin is not a non-polar matrix and we regret if this was not conveyed adequately. We have revised the manuscript accordingly. We refer to the cutin matrix as amphiphilic (abstract, page 2), specify that the composition is a long fatty acid-based polyester (page 2), and state that we originally expected (but later disproved) that the water transport behavior of olive cuticles is mainly governed by the lipophilic waxes and the hydrophilic polysaccharides (page 3). We thank the referee for the assessment that the artificial membranes are useful to understanding better asymmetric membranes, including cuticles. As suggested by the reviewer, we further compare the biological and artificial systems and highlight their intrinsic differences (Pages 10, 12), also taking into account new dynamic mechanical analysis (DMA) data acquired for the cuticles (vide infra).

It is difficult for a reader who is not in the cuticle field to understand the schematic diagram of the cuticle in comparison to the structure of the olive cuticle next to it. The olive cuticle has a particular complex multilayered structure. The olive cuticle structure would need much more explanation, i.e. annotation in the picture, in the main figure and extended data set. In addition, it is puzzling that the authors show different structures of the olive leaf cuticle without mentioning the frequency with which they were observed as well as conditions or developmental stage, etc. when they appear in the extended data set.

We thank the referee for this pertinent comment. We have carried out additional electron microscopy experiments and included these in the revised manuscript (revised Fig. 1a, new and revised Fig. S1c-f, new Fig. S10d-e, new Fig. S11b, new Fig. S13). Additional explanations were added to all electron microscopy images and the schematic of the cuticle cross-section was redrawn and re-labelled (revised Fig. 1b). We also conducted TEM image analysis to further examine the frequency of the observed structures in olive cuticles and discuss their morphology in detail (Fig. S1c-f, discussion in the new SI section S1.13 - pages 15-17).

For this reason, the olive leaf cuticle may not be the best suitable cuticle in comparison to the two-component system of the PG-membrane. Why did the authors not chose a simpler cuticle, such as the tomato fruit cuticle, having a uniform structure somewhat comparable to schematic diagram (except that the wax is not crystalline) and of which the composition/structure is also much better understood. Studying the complex olive leaf cuticle raises then the question whether the complex ultrastructure of which the chemical-structural basis is not known may be, at least in part, the source of the particular water transport behavior.

We selected olive cuticles because of (i) their strong transversal compositional heterogeneity (type 1-3 cuticles following Holloway's classification), (ii) their well-investigated chemical structure, and (iii) dense membrane transport properties (i.e. purely diffusional transport). They are also easier to handle than other cuticles due to their relatively large thickness, and this was crucial for some of the experiments carried out. We have included our rationale for the selection of olive cuticles in the revised manuscript (page 3, page 6). Indeed, in hindsight, ivy cuticles would have been a better choice (see below). For the reasons mentioned by the reviewer, tomato cuticles were in fact included in our initial screening of astomatous cuticular membranes that might display directional transport characteristics, but we excluded them from further studies, because these Type 4 membranes (all reticulate) display a more symmetric architecture than the olive and ivy cuticles that were selected.

Therefore, it would be useful to also include data of a simple-structured cuticle to show whether it behaves as the olive leaf cuticle under identical settings.

In response to this pertinent comment, we carried out additional experiments with ivy cuticles, which exhibit similar features as olive cuticles (i.e. strongly compositionally graded, dense membranes of well-known composition and structure), but they are also trichome-free (see reviewer 2 comments) and their cuticular pegs are less pronounced (new main text reference 52). We mention the ivy cuticles prominently in the abstract, on page 3, and discuss the new data (new Figures S11, S12, and new Table 6) in an extensive paragraph on page 9, and in the concluding section (pages 12-13, main text), as well as in the SI sections S1.12 (page 15, SI) and S1.13 (page 16, SI). Gratifyingly, the ivy cuticles show similar transport properties as the originally reported olive cuticles (Fig. 3 olives; Fig. S12a-b ivy), including directional transport behavior that results from a plasticizing effect, as supported by transport experiments and dynamic mechanical analysis (DMA) measurements (Fig. S12). Importantly, ivy cuticles do not exhibit pronounced inner cuticular side pegs, which complicate the data analysis in the case of olive cuticles (page 9 main text, see also SI section S1.12 – pages 14-15 of the SI). Thus, no normalization is necessary and the data clearly show directional transport for the wax-containing ivy cuticles (Fig. S.12a-b). Further details are provided in response to related comments made by referee 2.

Minor comments

A more detailed discussion with additional schematic diagrams explaining the cuticle as a two/multi component system of a polar and an amphipolar polymer both having (different?) swelling properties in contrast to the PG-membrane of which the apolar part does not have swelling properties would be useful. Of course, best would be to generate an artificial membrane having such properties and add water transport data of it. But this may be for future research.

We thank the reviewer for their suggestions. We have revised all schematics shown and hope they are now more clear. We also added water uptake data for olive cuticles (discussed on page 12 of the main text, see also SI section S.1.8 - page 10 of the SI) and have better explained their

response to hydration by using DMA measurements (Fig. 4, Table S7, main text pages 10-12, see also response to a comment made by referee 2).

Figure 1b; water drop would need explanation in legend or remove droplet. It is irritating as the cutin matrix of the olive leaf cuticle has also a round structure on the surface, that has nothing to do with the water drop.

To avoid any confusion, we replaced the schematic of the cuticle shown in Fig. 1b. The water droplet has been omitted.

Figure 1f and 1g as well as in extended data: Orange arrows are not explained in legend.

In all figure legends (including SI Figures), we have added explanations of the orange arrows that are used as a guide to the eye.

Furthermore, explain the role of waxes in barrier properties more precisely. Maybe wax influences also the swelling properties of the cutin polymer?

We have conducted water uptake experiments using olive membranes with (CM) and without (MX) waxes and observed no significant differences on the total water uptake (see main text page 12 and SI section S1.8 – page 10 of the SI). We have also added a short comment on the well-known barrier role of the waxes on water transport (page 8) and directed the readers to previous works focusing further on this aspect (reference 16).

Reviewer #2

Kamtsikakis and coworkers analyze the asymmetric diffusion of water across composite layers such as the olive leaf cuticle and bio-inspired SBS/CNC membranes. While the manuscript address an interesting and important question, I am concerned that some of the conclusions for the cuticle layer are drawn with limited structural analysis and experimental support.

We thank the reviewer for their time to review our manuscript, the assessment that it addresses an interesting and important question, and the valuable critique that has helped us to improve the paper.

1) The main text describes rather well the experiments made to ascertain the structure of the SBS/CNC membranes. In contrast, the text and supplementary material are much weaker when it comes to the structural description of the cuticle. I believe, the following issues will have to be addressed:

We thank the reviewer for their positive comment regarding the characterization of the SBS/CNC membranes and the suggestion to include more structural information on the biological membranes. We have now added a detailed supplementary section on the structure of olive cuticles (ATR-IR analysis, electron microscopy studies, see new SI section S1.13 – pages 15-17 of the SI), which further contributes to the already existing knowledge on the structure of olive cuticles. We have also added several references (new references 25 – 30, new SI references 27 – 31, and 35 – 41) to previous works focusing on the structure and composition of these membranes.

(a) I am unable to infer what I am seeing in Fig. 1a. To me, it looks as if a wax/cutin layer (above) has detached for the outer wall of an epidermal cell (below). Also, the authors have not labeled the layers that are visible and attempted to draw a parallel with the diagram of Fig. 1b. In the supplementary material, the authors provide more information on the structure of the cuticle but again it is very difficult to know what was, in concrete terms, the structure of the cuticle used in their experiments.

The referee's comment is well taken. We replaced the initial FIB-SEM image shown in Fig. 1a with a representative TEM image and added additional TEM images to the SI (new SI Fig. 1c-d), and extended our characterization of the architecture around trichomes (new Fig. S13). Following the referee's suggestions, which mirror comments of referee 1, we have labelled the EM images better (Fig. 1a, Fig. S1c-f, Fig. S11b) and added a new section (SI section S1.13, pages 15 – 17

of the SI), in which we discuss the structural morphology of olive cuticles in detail and compare our findings to results of previous reports (new SI references 28-29).

(b) Extended data Fig. 1 makes reference to two types cuticles: Type 1 (lamellate/reticulate) and Type 2-3 (faintly lamellate, amorphous and gradually reticulate architecture). Do the author know what was the dominant type for their cuticle samples? What are the relative areas occupied by these cuticle types in the tested samples? There is a striking difference in the thickness of the outer lamellate layer between Type 1 and Type 2-3. Does this structural difference, or any other structural differences between the types, have a functional effect on the diffusion of water?

In new experiments, we used TEM to image the cross-sections of olive cuticles (new Fig. S1c-d, new Fig. S13 focusing on trichomes, discussion in new SI section S1.13 – pages 15 - 17 of the SI) and ivy cuticles (Fig. S.11b). In the case of olive cuticles, less than 5% of the total area (determined with TEM) is occupied by lamellate regions and therefore the dominant type is the lamella-free morphology, as previously reported (new SI references 28-29). Since these lamellate regions are structurally ordered, they *might* restrict locally mass transport, but since they occupy only a small fraction of the surface area their contribution should be negligible.

(c) There are no stated attempts to characterize the integrity of the isolated cuticles. Since the cuticle has low conductivity to water, even small, hard-to-notice, defects could easily govern the diffusion of water. Along the same line, how does the thickness of the different layers vary spatially? Again, reduction in thickness of the water impermeable layers, even over small areas, could easily control the overall behavior of the cuticle sample.

We thank the reviewer for pointing out this important aspect. Indeed, we used a well-known procedure (“ethanol test”) (added main text reference 71) to characterize the integrity of the enzymatically isolated cuticles before using them for the permeation tests. This standard experimental procedure was not specifically mentioned in the original manuscript, but is now mentioned in the experimental section (page 18). The observed mass transport properties of olive cuticles are also indicative of a dense material demonstrating purely diffusional transport characteristics in agreement with previous works (new references 27 and 30, see main text page 3 and discussion in SI section S1.13 – page 17 of the SI). We have also added comments on the thickness variation of olive cuticles (see main text page 6 and SI section S1.3 – pages 6 – 7 of the SI). The thickness of the trichome-free areas is relatively homogeneous (e.g. for CM: 7 ± 1 μm , MX: 8 ± 1 μm , $n = 15$ using 5 different TEM images) (see also SI S1.3). However, trichomes “inflate” locally the observed cuticular thickness (SI S.1.3, Fig. S13). This local variability of the thickness also leads to a systematic underestimation of the storage modulus values by overestimating the cross-sectional area to calculate the modulus as previously highlighted (new SI reference 5, see SI section S.1.3 and main text page 6). A cuticle of similar thickness to the one present in the trichome-free areas is also covering underneath the olive trichomes (Fig. S13).

(d) The chemical characterization of the cuticle is minimal and none of it is presented in the figures of the main text. I believe additional characterization would be useful. For example, I would have liked to see staining for specific polysaccharides (cellulose, pectins) and waxes. According to the authors, the cuticle is, on average, 20 μm thick. Therefore, it should be possible to resolve the distribution of some chemical components both in the plane of the cuticle AND across its thickness.

We thank the reviewer for their suggestion to include more structural information about olive cuticles. In view of the previously established chemical and structural composition of olive cuticles, it was initially out of scope of this work to extensively study their structure. To demonstrate the transversal compositional gradient of the materials, we have added a new dedicated section (see SI section S1.13 – pages 15 – 17 of the SI) discussing the ATR-IR spectroscopy and electron microscopy (FIB-SEM, TEM) data (Fig. S1, Fig. S13), which further contributes to the already existing knowledge on their structure (new SI references 28 – 29) and composition (new SI references 27, 30, 31). We have also added a figure (Fig. S13) of the olive cuticle morphology focusing around trichomes, which is discussed in the same SI section (SI S1.13 – page 17 of the SI). The same characterization methods (ATR-IR spectroscopy, TEM analysis – Fig. S11) were

applied for ivy cuticles to demonstrate their well-known transversal heterogeneity. For space reasons and in view of their well-known structure from literature (new SI reference 24), the respective ivy structural data of Fig. S11 are not discussed in detail in the SI section S1.13 – page 16 of the SI).

(e) Finally, why are the authors studying the cuticle of Olea specifically? The authors do not state evidence that the cuticle of this species has some special properties. Do they have reasons to believe that Olea has a cuticle better adapted for asymmetry diffusion of water? It seems strange to talk about bio-inspired membranes from Olea if the species is not unique in some way.

As pointed out in response to a similar question/comment made by referee 1, we selected olive cuticles because of (i) their strong transversal compositional heterogeneity (type 1-3 cuticles following Holloway's classification), (ii) their well-investigated chemical structure, and (iii) dense membrane transport properties (i.e. purely diffusional transport). They are also easier to handle than other cuticles due to their relatively large thickness, and this was crucial for some of the experiments carried out. We have highlighted the primary reasoning for the selection of olive cuticles in the revised manuscript (page 3, page 6).

2) The authors did not mention that the adaxial leaf surface of Olea has trichomes. Some of them are seen in Ext. Data Fig. 1g,h. What was the density of these trichomes? Can the authors confirm that these trichomes do not function as "water pores" in isolated cuticles? Trichomes of xerophytes have been shown to transport water asymmetrically, favoring the inward flow of water as observed in their work

The referee raises a good point. We have revised the manuscript to clearly bring out that the olive cuticles contain trichomes (pages 6 - 7). Based on new SEM data (discussion in new SI section S1.13, new Fig. S13), the trichomes cover about 18% of the surface (SI section S1.13, page 17 of the SI). We included new transmission electron microscopy images (new Fig. S13), which show that the trichomes of olives cuticles are not "leaky pores", since the dense cuticular layer, which dictates mass transport and ultimately the observed directionality, extends underneath the trichomes. We mention this explicitly on page 7 of the revised manuscript. Finally, the original transport data (Fig. 3) show clearly that the membranes under investigation are non-porous, as also supported by previous works (new references 27 and 30) (see discussion in SI section S1.13, page 17 of the SI).

3) The authors use the results of Fig. 3 to claim that diffusion across the cuticle is asymmetric when the relative humidity is low (2%) on the receiver side. However, this conclusion is reached only when the data are corrected for a postulated effect of excess area on the ICS. Looking at the RAW data measured from the UNTREATED cuticle (i.e. the closed, uncorrected, symbols in Fig. 3a,b), one sees that the measured permeance is not affected by the orientation of the cuticle at RH = 2% and only slightly so for RH = 100%. A statistical analysis of the effect of membrane orientation on the permeance would almost certainly show that the RAW distributions are not statistically different. Unfortunately, such analyses were not done. Instead, the authors cite other studies that have claimed an effect of surface roughness on the permeance. Based on these studies, the initial distributions are rescaled when the ICS is on the donor side. It is only after this correction is made that the authors are able to make the claim mentioned above. The conclusion reached by the authors hinged 100% on their hypothesis that the raw data must be rescaled to take into account the ICS roughness. Although it is quite possible that surface roughness, in some context, can affect permeance, I believe the authors are obligated to show that surface roughness is important for the intact cuticle of Olea. Without this experimental support, the authors have effectively taken two distributions that are not statistically different and made them different without much more than a hunch that it ought to be so.

The reviewer is correct that the fact that asymmetric transport in wax-containing olive cuticles (CM) is only apparent after correcting for the ICS roughness is really a fly in the ointment. However, we respectfully maintain that the experimental studies that we cited and sorption theory paint a very clear picture and show that it would be misleading *not to apply* a correction for the geometric increase of the sorption area. The referee's point that it should be shown that surface

roughness is important for the intact (i.e., the wax-containing) cuticle is well taken, but this is experimentally impossible, as the enzymatic isolation process of olive cuticles cannot remove the pronounced cuticular pegs (i.e. cell wall protrusions), which are present in the inner cuticular side of the membranes (Fig. S10, SI section S1.12 – pages 14 – 15 of the SI). Moreover, chemical or mechanical treatment of the isolated cuticles damages their fragile structure (SI section S1.12 – page 14 of the SI). To address this problem, we carried out additional transport experiments with astomatous, enzymatically isolated **ivy** cuticles (new Fig. S12, Table S6), which have a similar compositionally asymmetric architecture as the *olea* cuticles (new Fig. S11), but the cuticular pegs are much less pronounced (new Fig. S10d-e) so that no correction for the increased surface area must be carried out. Gratifyingly, the ivy data collected under dry receiver conditions show highly directional transport from the outside to the inside, with a *PAF* value of 0.60 (new Fig. S12, Table S6), which is very similar to the (surface-area corrected) value observed for the olive membranes (0.62; Table S3) and (in view of the similarities of the employed cuticles) supports the conclusion that the correction is adequate. **In any case, unequivocal proof for directional transport in ivy cuticles was obtained.** We discuss the new ivy data on page 9 of the revised manuscript. We elected to maintain the narrative of the paper centered around olive cuticles (since more exhaustive data are available), but should the referees take any issues with this approach, we would be happy to move the ivy data to the center stage. We edited our wording throughout; for example, in the introduction we state that “the transport of water through astomatous olive and ivy leaf cuticles can indeed be directional...”, reflecting that the extent of directionality depends on the cuticle type, relative humidity, and corrections applied.

Based on these three comments, I believe the central claim that “transport of water through astomatous olive leaf cuticles is indeed directional and that the permeation is regulated by the hydration level of the cutin-rich layer” is not properly established.

We trust that the first element of the referees’ concern, i.e., that “transport of water through astomatous olive leaf cuticles is indeed directional” has been addressed by the addition and corrections detailed above. We further addressed the second element, i.e., the fact “that the permeation is regulated by the hydration level of the cutin-rich layer is not properly established” by conducting dynamic mechanical analysis (DMA) measurements of wax-containing and wax-free olive (Fig. 4) and ivy cuticles (Fig. S12) under ambient (50% RH) and wet conditions. The data, which are discussed in new text on pages 10-12, have afforded strikingly detailed DMA traces of such cuticles. The data show clearly that the cuticles are plasticized by water (Fig. 4c) and this plasticization mechanism is also reflected by the humidity-dependent permeation properties of the outer cuticular side upon hydration (Fig. 3a), which effectively leads to increased water permeability and directional transport properties as expected from polymer membrane theory. The same plasticizing effect is present in ivy cuticles (Fig. S12).

Minor comments

1) line 26: “Directional transport is ubiquitous in natural organisms, which use this mechanism for water collection and retention” This statement is supported by citing ref(11), yet the cited paper is about the directional migration of droplets on spider silk, which is very different from the meaning given to “directional transport” in the remainder of the paper. Also, to my knowledge, spiders do not use this mechanism “for water collection and retention”. Please cite a more relevant paper.

We have replaced the originally cited reference with the following new references, which are more relevant to plants and cover both transport mechanisms (i.e. convective, diffusional) and directionalities (i.e. lateral, transversal direction): (1) J. Ju et al., A multi-structural and multi-functional integrated fog collection system in cactus. *Nature Communications* 3, 1247 (2012). (2) P. S. Raux, S. Gravelle, J. Dumais, Design of a unidirectional water valve in *Tillandsia*. *Nature Communications* 11, 396 (2020).

2) line 222: “Our data suggest that the water losses of plants through the cuticles double in wet conditions, which is ecologically meaningful.” I believe this statement is incorrect. The permeability to water, not the water losses, is doubled under wet conditions. Presumably, the

effect on water losses is negligible since the environment is wet and therefore the gradient for water diffusion is weak.

We thank the reviewer for noting this. We have modified this conclusion statement (page 12).

3) The authors rely heavily on acronyms. I believe the text would be easier to follow if fewer acronyms were used.

For space reasons, we have used standard acronyms employed in polymer (e.g. SBS, CNCs) and cuticular literature (e.g. CM, MX). If needed, we can add an abbreviation table to assist with the reading.

4) It is good that the authors are using a standard protocol to measure the water permeability of the membranes. However, the authors refer to “wet cup” and “dry cup” set-ups in the main text without explaining clearly what they are. Perhaps it would be useful to add the diagrams of Extended Data Fig. 8 a,b to the figures in the main text so that a reader unfamiliar with this type of experiments could understand what was done. Also, I could not find curves of cup weight vs time which were used to compute the permeance. It would be good to show some of these curves so that the reader can verify the linearity of the gravimetric method.

For space reasons we did not discuss these well-established transport methods further in the main text. However, a detailed description of these methods can be found in the SI section S1.2 of the revised manuscript. Following the reviewers' suggestions, we have redesigned the supplementary figures associated with the gravimetric cup methods and have added some typical curves to allow verification of the linearity of the technique as suggested (new Fig. S8a-b).

5) line 186: “While our data show that the PAF depends considerably on the ICS surface area employed to calculate the permeance, the wax-free (MX) cuticles (Fig. 3c) display such a pronounced directionality that irrespective of the data analysis, the preferred water transport direction is unequivocally from the outside to the inside.” I believe the first part of this statement is misleading. The authors have not demonstrated with their data that surface roughness is relevant for permeance or PAF measurements. For their data to SHOW dependence, they would have had to perform experiments where surface roughness is varied and its effect on permeance measured. The authors have not done such experiments. Instead, they assume that surface roughness might be important (based on publications on other systems) and analyze their dataset in two different ways. Please clarify the first part of this statement to avoid confusion

To avoid confusion, we have rephrased the first part of the statement as follows:

“While the absolute value of PAF **varies** considerably on the ICS surface area employed to calculate the permeance, the wax-free (MX) cuticles (Fig. 3c) display such a pronounced directionality that irrespective of the data analysis, the preferred water transport direction is unequivocally from the outside to the inside.”

Of course, experiments in which the surface roughness is varied would be extremely valuable, but since biological species are investigated, such variation is unfortunately impossible (no further edits). We trust that the new experiments featuring peg-free ivy cuticles resolves this issue.

Other Changes

1. The abstract has been shortened and the references in the abstract have been removed to meet the formatting guidelines of *Nature Communications*.
2. All schematics have been redesigned to improve clarity and visual appeal.
3. All main text figures (Fig. 1 – 4) along with their captions have been placed after the references of the main text.
4. The references of the main text and the references of the experimental section have been merged into one section.

5. Several references were added in which we cite relevant fundamental studies on the architecture and composition of olive membranes (new references 25 – 30, SI references 27 – 31, and 35 – 41), as well as works highlighting the phase transitions of cuticular membranes and the plasticizing effect of water on cuticles (new references 55 – 59).
6. The “Extended data figures” were transferred to the Supplementary Information.
7. For space reasons, some experimental methods were transferred to the SI, including electron microscopy studies (now SI section S1.1), the ASTM E96 method (now SI section S1.2), and the DMA procedure (SI section S1.3).
8. After the modifications in the text, the order of the SI headings and figures has been changed. All cross-references have been updated.
9. A new section about the transversal heterogeneity of olive cuticles has been added in the SI (section S1.13).
10. Some FIB-SEM images have been replaced and annotations have been added (Fig. S1), images of the inner cuticular side of ivy cuticles have been added (Fig. S10), and additional figures with data about the ivy cuticles were also added to the SI (Fig. S11-12). EM images focusing at the trichomes of olive cuticles have been added (Fig. S13).
11. Tables with DMA data on olive cuticles (Table S7) and THO transport measurements (Table S6) with ivy cuticles have been added.

REVIEWER COMMENTS

Reviewer #1 (Remarks to the Author):

Dear authors,

You have carefully revised the manuscript and included most of the recommendations of the reviewers. The better characterization of the olive cuticle as well as the addition of investigations of a second type of a cuticle having a dense outer layer, the ivy cuticle, were very valuable. Additional investigations of the olive and ivy cuticles revealed changes of the properties of the cuticle depending on the humidity, i.e. demonstrating a plasticizing effect of water and explaining very well the findings. The findings are a great advance to our understanding of plant cuticles. The generation of artificial cuticle-inspired membranes have greatly helped in the understanding of such membranes and the lessons learnt from the cuticle may help to produce material with novel features in the future.

Unfortunately, you did not react to the comment of one of the reviewers that the number of abbreviations used in the text should be reduced. I would like to fully support my colleague in this statement. This paper is particularly difficult to understand because of the high number of abbreviations and acronyms (close to 30 in the main body of the text), which is a pity for such an interesting paper. Indeed, there are many abbreviations standard and useful, but others are particular to this paper or only used by a small group of scientists. Thus, it is not necessary to introduce a broad readership to all of these abbreviations.

Some of the abbreviations are also odd, for example the most commonly used opposite of "upper" is "lower" and from "bottom" is "top", thus, the combination upper side (US) and bottom side (BS) is not intuitive. Since "S" is already used as abbreviation for "styrene" the use for "side" is a duplication and irritating. With the respective description in the Figure legends, a self-understandable labeling in the Figures with "top" and "bottom" should be used throughout the paper. Then, no abbreviations are necessary.

Similar, the cuticle nomenclature could be simplified. Sometimes the meaning is also doubled, such as wax-containing cuticle (CM) or cuticle without wax (MX). The abbreviations are irritating and not necessary. The cuticle nomenclature paper would be not only easier to understand, but also more precise when uniformly "native cuticle" or "waxy cuticle" is used for the cuticle that is not wax-extracted and "dewaxed cuticle" for the extracted one, without any abbreviations.

"TOH" for "tritiated water" is not self-understandable, while "3H₂O" would be. Also, simply "H₂O" could be used in the figures or formulas and only in the legend/methods and it could be explained that tritium-labeled was used for the assay.

The abbreviations "PAF" and "RCC" could be used in the Figures only and described in the Figure legends, but no abbreviation should be used in the main text since abbreviations are much further away from their explanation. A text explanation for $\tan\delta$ is missing.

At least in Biology "vide supra" and "vide infra" are very unusual expressions, please say simply "see above" and "see below".

Thus, please revise the text and the figures that they are easier understandable for a broad readership.

Other comments:

Line: 237: Citation of Figure 4: No data on ivy, add appropriate Supplemental Figure citation.

Lines 240: Range -35-35. Correct?

The information that only cuticles having a particular dense outer layer, such as these from olive and ivy leaves, were used, i.e. could be used for these studies should be added and discussed.

Figure 1

a and b) The double-headed arrows labeling the schematic diagrams do not label the different layers accurately. The symbol used in the Figure S13 is much better. The symbols should be accurately placed accordingly to size of the layers at the right border of the image to allow easier interpretation. If you would like to indicate that cutin is also in the cell wall layer (as the arrow indicates) then the brown colored zone would need to have a black gradient color fading towards the bottom.

e-g) Orange arrows as simple "guides for the eye" should be removed. An arrow head at the end of the curve in a graph is irritating. However, potentially an orange arrow indicating the "gradient from higher to lower cellulose content" could remain in the Figures panel showing composite membranes. Then the arrow needs only to be removed from the panel "Neat SBS". Similar adaptations are necessary for Figures S6 and S7.

Since in several Figure panels mathematical equations are indicated a "hyphen" can be misinterpreted as "minus", better use a colon instead since you use a slash as symbol for division. Similar, also in other panels, such as in Figure 2.

"I" is not explained in the legend.

Figure 2

e) Main text may not fit to Figure: 74% increase for 33 μm membranes, but not 206% for 120 μm . It could be even higher or just difficult to read? Curves in all panels as in panel f would be helpful, but the curve should best have the color of the symbols (or the meaning of the color should be explained).

Figure 3)

See comments above in respect to THO, CM and MX. Explain PAF in legend. Maybe add "under different humidity conditions" in the title/first phrase of Figure legend.

Figure 4) See above comments to cuticle terminology, E' is missing for storage modulus at the Y-Axis as well as text expression of "tan delta"(Dissipation factor ?). Apply also for all other Figures.

Supplemental Figures:

Figure S1, see comment on double headed arrows and its localization (Figure 1). Harmonize with text to FIB-SEM, instead of FIB/SEM. Remove orange arrows or change its meaning (see above). Revise cuticle nomenclature (see above).

Figure S3: Are the pictures of scintillation vials necessary?

See above for "Top" and "bottom" nomenclature (no abbreviation necessary), improve nomenclature also in Figures S4, S5, S6, S7. Orange arrows: See above for Figure 1.

Figure S8:

a,b) Render the equations placed in the graphs more understandable. Maybe place SBS/CNC 15% in the title of the graphs, here several meanings of "%" and of "=".

Figure S10:

a) What represents the right double-headed arrow?

Figures S9, S12: What expresses $\tan\delta$? What does the arrows in Figure 12 c represent?

Figure S13: c) Maybe you indicate the locally milled area with an arrow? ETD means Everhart-Thornley detector?

I hope that these comments will be helpful to revise the manuscript.

Reviewer #2 (Remarks to the Author):

I thank the authors for their response to my comments. However, I still have some comments and concerns, which I'll present using the figures to order my thoughts.

1. Fig. 1 has been improved. It is now easier to understand the structure of the cuticle.

2. I noted a few minor issues with Fig. 2.

2.1. In the various panels of Fig. 2, the authors manipulate three control parameters: concentration of CNC, membrane thickness, and RH. Typically, one of the control parameters is kept constant while the others are varied. However, it is not always easy to know the value of the control parameter that is kept CONSTANT. For example, a careful reading of the figure legend is necessary to find out the value of the membrane thickness ($33 \pm 5 \mu\text{m}$) in Fig. 2C. Therefore, I would suggest the put the value of the thickness in the panel itself in Fig. 2a-d so that all three control parameters are quickly accessible. On a similar note, what is the RH in Fig. 2F? Please, include it within the panel.

2.2. The authors present the permeability of the membranes, an intrinsic property of the material. Based on the methods, the latter is calculated using the standardized protocol put forward in ASTM M95. In my version of ASTM M95, it is stated on page 5 that: "the calculation of permeability is optional and can be done only when the test specimen is homogeneous (not laminated) and not less than 1/2 in. [12.5 mm] thick ...". It seems the SBS/CNC membranes fail on both counts. For example, the thickest of the membranes used in the paper is 100 times thinner than the minimum thickness recommended in the international standard. Can the authors give a justification?

2.3. In Fig. 2 c and f, does "a.u." mean "arbitrary unit"? The permeability asymmetry is a non-dimensional ratio. It has no unit.

2.4. The authors never clearly explain why permeability increases with thickness for the SBS/CNC 15% membranes. They simply say "but a dependence on the thickness has been reported for hydrophilic membranes and this was explained with swelling effects". However, to my knowledge, this is valid for materials with high sorption capacity. Their SBS/CNC membranes at best can absorb 4% water (Fig. 2d) which would suggest that swelling is minimal.

3. Regarding the data reported in Fig. 3, I appreciate greatly the additional work performed by the authors to establish more firmly the transport asymmetry in intact cuticles. I believe the way they chose the present the new data, i.e. as suppl. mat., works well. Note, however, that in Fig. S12 a, the upper labels should read "Outward tranSport" and "Inward tranSport".

4. I am completely baffled by the mechanical analysis of Fig. 4 and the authors' attempt to extract from it an explanation for the water transport asymmetry in leaf cuticles. I feel very uncomfortable with the inferences that are made based on these results. First, as the authors noted themselves, the cuticle is a complex material made of at least three distinct compounds (waxes, cutin, polysaccharides) and its structure is both heterogenous and anisotropic. Given this complexity, the interpretation of water transport experiments and mechanical experiments must be done very carefully. Normally, this is done by isolating every single component to understand its behavior. While the authors have made some attempts to "simplify" their cuticles by removing waxes, they are still dealing with a complex material. Therefore, the suggestion that plasticization of cutin explains the transport asymmetry is, in my view, only one of many other possible explanations. It seems the idea is put forward only because it allows the authors to reconcile their results for the SBS/CNC membranes with those for the cuticle. The results presented in fig. 4 certainly do not provide solid evidence that changes in the cutin layer are the only realistic mechanism for the water transport asymmetry. Second, the transport asymmetry measured by the authors is small (not even a factor of 2). Therefore, minor changes in any of the components of the cuticle could account for the observed asymmetry. It is not clear to me that the change must be of a magnitude that can be picked up by a mechanical test and the structural change certainly does not need to be associated with a phase transition.

Reviewer #3 (Remarks to the Author):

The paper entitled "Asymmetric water transport in olive cuticles and cuticle-inspired, compositionally graded membranes" by Kamtsikakis et al. is an interesting and timely work about the water transpiration mechanisms through plant cuticles, in particular those from olive and ivy leaves. The authors claim that the mechanism depends on the humidity (the hydration of cutin polymer) and support their results by using a model "inspired" by plant cuticles. Such a model can find applications in different industrial uses (i.e. smart packaging). The text is academic and easy to follow and figures are intuitive and very nice (especially 3d models). The experimental section is well described, providing details enough to be reproduced.

I have some concerns and recommendations that should be addressed before publication:

- 1.- In my opinion, "ivy cuticles" should be also included in the title.
- 2.- I miss a good comparative (maybe a table or something similar) between authors' model and other cuticle-mimetic systems, indicating their different chemical compositions.
- 3.- The participation of waxes, that are essential in the water transport of plant cuticles, is absent in authors' model. Practically a three-layer system (polysaccharides/cutin/epicuticular waxes) is compared to a two-layer membrane (CNC/SBS). Please, devote some lines to this during discussion.
- 4.- Other important aspect for water permeability in polymers is crystallinity. The authors explain well this characteristic for CNC particles, but what about SBS? Is also this polymer crystalline or not? Please, consider that cutin is an amorphous biopolymer, while in the case of waxes the situation is more complex, although they tend to be found as crystals.
- 5.- In the conclusions, the authors suggest to change CNC by PVA. Although this modification is correctly reasoned, it could be interesting for the industrial application of membrane, but not for cuticle-inspired systems. Maybe, the use of polysaccharide-based fibers is a best choice for the latter case.
- 6.- The chemical differences between SBS and cutin are very important. Among them, SBS is a

linear polymer (a thermoplastic), while cutin shows a complex behavior typical of thermosets with no melting point and lack of solubility. Can also these differences influence the water permeation?

Response to Reviews and Reference to Changes (NCOMMS-20-18275A-Z)

Reviewer #1

Dear authors,

You have carefully revised the manuscript and included most of the recommendations of the reviewers. The better characterization of the olive cuticle as well as the addition of investigations of a second type of a cuticle having a dense outer layer, the ivy cuticle, were very valuable. Additional investigations of the olive and ivy cuticles revealed changes of the properties of the cuticle depending on the humidity, i.e. demonstrating a plasticizing effect of water and explaining very well the findings. The findings are a great advance to our understanding of plant cuticles. The generation of artificial cuticle-inspired membranes have greatly helped in the understanding of such membranes and the lessons learnt from the cuticle may help to produce material with novel features in the future.

We thank the referee for their time to re-evaluate this manuscript, the positive assessment, and their recommendations, which have helped us to improve this manuscript.

Unfortunately, you did not react to the comment of one of the reviewers that the number of abbreviations used in the text should be reduced. I would like to fully support my colleague in this statement. This paper is particularly difficult to understand because of the high number of abbreviations and acronyms (close to 30 in the main body of the text), which is a pity for such an interesting paper. Indeed, there are many abbreviations standard and useful, but others are particular to this paper or only used by a small group of scientists. Thus, it is not necessary to introduce a broad readership to all of these abbreviations.

We apologize if we did not pay sufficient attention to this point. In response to the reviewers' comments, we have removed many of the previously used abbreviations and hope that the manuscript has become more accessible (see comments below for point-by-point changes).

Some of the abbreviations are also odd, for example the most commonly used opposite of "upper" is "lower" and from "bottom" is "top", thus, the combination upper side (US) and bottom side (BS) is not intuitive. Since "S" is already used as abbreviation for "styrene" the use for "side" is a duplication and irritating. With the respective description in the Figure legends, a self-understandable labeling in the Figures with "top" and "bottom" should be used throughout the paper. Then, no abbreviations are necessary.

We have replaced the abbreviations upper side ("US") and bottom side ("BS") with the terms "top side" and "bottom side" as suggested by the referee. These changes were applied to all relevant Figures and text.

Similar, the cuticle nomenclature could be simplified. Sometimes the meaning is also doubled, such as wax-containing cuticle (CM) or cuticle without wax (MX). The abbreviations are irritating and not necessary. The cuticle nomenclature paper would be not only easier to understand, but also more precise when uniformly "native cuticle" or "waxy cuticle" is used for the cuticle that is not wax-extracted and "dewaxed cuticle" for the extracted one, without any abbreviations.

We have removed the abbreviations "CM" and "MX" in the revised MS. We now use the terms "wax-containing" (previously "CM") and "wax-free" (previously "MX") cuticles throughout.

"TOH" for "tritiated water" is not self-understandable, while "3H2O" would be. Also, simply

“H₂O” could be used in the figures or formulas and only in the legend/methods and it could be explained that tritium-labeled was used for the assay.

As suggested by the referee, we have replaced “THO” with the self-explanatory term “³H₂O”.

The abbreviations “PAF” and “RCC” could be used in the Figures only and described in the Figure legends, but no abbreviation should be used in the main text since abbreviations are much further away from their explanation. A text explanation for $\tan\delta$ is missing.

We have also replaced the abbreviations “PAF” and “RCC” throughout the text and used the intuitive terms “inward” and “outward” transport for the orientation-dependent permeation experiments. We have further added a short description for $\tan\delta$ in the Materials & Methods section (SI Line 88).

At least in Biology “vide supra” and “vide infra” are very unusual expressions, please say simply “see above” and “see below”.

We have replaced the terms “vide supra” and “vide infra” with “see above” and “see below”, respectively.

Thus, please revise the text and the figures that they are easier understandable for a broad readership.

We thank the reviewer for their suggestions, which indeed have helped us to render the text more understandable to a broader audience. In addition to the changes detailed above, the acronyms ICS (i.e., inner cuticular side) and OCS (i.e., outer cuticular side) were replaced with the full terms in the main text and were kept only in the SI Figure S10 to explain other quantities that are needed to calculate the surface area of the inner cuticular side.

Other comments:

Line: 237: Citation of Figure 4: No data on ivy, add appropriate Supplemental Figure citation.

We have added a cross-reference to the ivy data (now line 230).

Lines 240: Range -35-35. Correct?

We have replaced the “-” with “and” in order to avoid confusion (now line 228).

The information that only cuticles having a particular dense outer layer, such as these from olive and ivy leaves, were used, i.e. could be used for these studies should be added and discussed.

We already stated that dense (i.e., non-porous) cuticles were selected (main text Lines 126-128 and 180-182) to “allow investigation of purely diffusional transport through dense matter” (no further edits).

Figure 1 a and b) The double-headed arrows labeling the schematic diagrams do not label the different layers accurately. The symbol used in the Figure S13 is much better. The symbols should be accurately placed accordingly to size of the layers at the right border of the image to allow easier interpretation. If you would like to indicate that cutin is also in the cell wall layer (as the arrow indicates) then the brown colored zone would need to have a black gradient color fading towards the bottom.

We elected to keep the arrows in Fig. 1, since they should indicate that the different pseudo-layers have no sharp boundaries, which is different from Fig. S13 (showing the cuticle as a whole). However, in response to the referee’s comment and to make the representation more realistic, we have made the cutin arrow in Fig. 1 shorter and moved it closer to the boundary with the cell wall layer.

e-g) Orange arrows as simple “guides for the eye” should be removed. An arrow head at the end of the curve in a graph is irritating. However, potentially an orange arrow indicating the “gradient from higher to lower cellulose content” could remain in the Figures panel showing composite membranes. Then the arrow needs only to be removed from the panel “Neat SBS”. Similar adaptations are necessary for Figures S6 and S7.

Based on the reviewer’s comment, we removed all the orange arrows from the Raman box charts and the SEM images in Fig. 1 and Supplementary Information Figs. S1, S4-S7 and S11.

Since in several Figure panels mathematical equations are indicated a “hyphen” can be misinterpreted as “minus”, better use a colon instead since you use a slash as symbol for division. Similar, also in other panels, such as in Figure 2.

“l” is not explained in the legend.

Based on the referee’s suggestion, but also the comments from referee #2, we have decided to reformat Fig. 2. We have added the self-explanatory schematics of the gravimetric cup method in Fig. 2a-b and accordingly modified the rest of the panels (Fig. 2c-f) (e.g., removal of “dry cup” and “wet cup” labels from the legend, addition of membrane thickness in the individual panels). For space reasons, Fig. 2 panels e,f were separated off into new Fig. 3a,b. We also edited the legends of Fig. 1 and Figs. S5 and S6 to provide an explanation of “l”.

Figure 2e) (current Fig. 3a) Main text may not fit to Figure: 74% increase for 33 μm membranes, but not 206% for 120 μm. It could be even higher or just difficult to read? Curves in all panels as in panel f (current Fig. 3b) would be helpful, but the curve should best have the color of the symbols (or the meaning of the color should explained).

We have double-checked our calculations and they are correct. When the **bottom** side of the SBS/CNC 15% membrane faced the donor, the water permeability increased from 3.16×10^{-14} to $9.68 \times 10^{-14} \text{ kg m m}^{-2} \text{ s}^{-1} \text{ Pa}^{-1}$ when the thickness was increased from ~ 33 to $\sim 120 \mu\text{m}$ and this corresponds to a 206% increase. By contrast, when the **top** side of the SBS/CNC 15% membrane faced the donor, the water permeability increased from 2.20×10^{-14} to $3.83 \times 10^{-14} \text{ kg m m}^{-2} \text{ s}^{-1} \text{ Pa}^{-1}$ when the thickness was increased from ~ 33 to $\sim 120 \mu\text{m}$ and this corresponds to a 74% increase.

In our opinion, adding curves would clutter the scatter graphs and therefore we decided not to include them, except in Fig. 3b, where the line is useful to reflect the linear relation between the PAF and the membrane thickness. We also changed the colors of the curves in Fig. 3b as requested.

Figure 3) (current Fig. 4) See comments above in respect to THO, CM and MX. Explain PAF in legend. Maybe add “under different humidity conditions” in the title/first phrase of Figure legend.

We edited the manuscript in line with these comments as detailed above.

Figure 4) (current Fig. 5) See above comments to cuticle terminology, E' is missing for storage modulus at the Y-Axis as well as text expression of “tan delta”(Dissipation factor ?). Apply also for all other Figures.

We added the “E” symbol in the y-axis of Fig. 5a,c, Fig. S9a,c and Fig. S12c,d. We have also added the term “damping factor” for $\tan\delta$ in Fig. 5b and Figs. S9b and S12c.

Supplemental Figures: Figure S1, see comment on double headed arrows and its localization (Figure 1). Harmonize with text to FIB-SEM, instead of FIB/SEM. Remove orange arrows or change its meaning (see above). Revise cuticle nomenclature (see above).

Please see response above regarding the use of double-headed arrows. The acronym “FIB-SEM” has been harmonized throughout, and abbreviations and orange arrows have been removed.

Figure S3: Are the pictures of scintillation vials necessary?

See above for “Top” and “bottom” nomenclature (no abbreviation necessary), improve nomenclature also in Figures S4, S5, S6, S7. Orange arrows: See above for Figure 1.

At their current image size and quality, the images of the vials do indeed not offer any substantial information and they were removed. We followed all other formatting suggestions as detailed above.

Figure S8: a,b) Render the equations placed in the graphs more understandable. Maybe place SBS/CNC 15% in the title of the graphs, here several meanings of “%” and of “=“.

We have edited the Supplementary Fig. S8 and partially moved the legends out of the graph, using semicolons, and adding the wt% symbol next to “CNC” to differentiate them better.

Figure S10: a) What represents the right double-headed arrow?

The right double-headed arrow was used to denote the thickness of the membrane. Since it was not a crucial piece of information and its meaning was apparently confusing, it was removed.

Figures S9, S12: What expresses $\tan\delta$? What does the arrows in Figure 12 c represent?

We have revised the manuscript and now explain E' and $\tan\delta$ in main Fig. 5 (see also comment above). The arrows in Fig. S12c were used to indicate to which axis the curves correspond to, in order to make the figure also readable if presented in black/white. After printing it, it appears that the curves can be differentiated and thus the arrows were removed.

Figure S13: c) Maybe you indicate the locally milled area with an arrow? ETD means Everhart-Thornley detector?

We think that the locally milled area is self-explanatory (no further edits). We have replaced the abbreviation ETD with Everhart-Thornley detector.

I hope that these comments will be helpful to revise the manuscript.

We sincerely thank the referee again for their valuable comments, which have helped significantly in improving the quality of this work.

Reviewer #2

I thank the authors for their response to my comments. However, I still have some comments and concerns, which I'll present using the figures to order my thoughts.

We thank the referee for their time to re-evaluate this manuscript and their recommendations, which have helped us to improve this manuscript.

1. Fig. 1 has been improved. It is now easier to understand the structure of the cuticle.

We thank the referee for their positive comment. We hope that our extensive changes (to Fig. 1 and most other Figures, see above) have addressed the referees concerns.

2. I noted a few minor issues with Fig. 2.

Following the comments from all reviewers, we decided to add schematics illustrating the gravimetric cup experiments to Fig. 2 (panels a-b). We trust that the illustrations make it easier for the reader to appreciate the different parameters and geometries discussed. For space reasons, the data related to the effect of membrane thickness were moved to a new main text figure (current Fig. 3). The figures and the captions have been edited accordingly (see also reply to comments from Reviewer #1).

2.1. In the various panels of Fig. 2 (current Fig. 3), the authors manipulate three control parameters: concentration of CNC, membrane thickness, and RH. Typically, one of the control parameters is kept constant while the others are varied. However, it is not always easy to know the value of the control parameter that is kept CONSTANT. For example, a careful reading of the figure legend is necessary to find out the value of the membrane thickness ($33 \pm 5 \mu\text{m}$) in Fig. 2C (current Fig. 2e). Therefore, I would suggest the put the value of the thickness in the panel itself in Fig. 2a-d (current Fig. 2c-f) so that all three control parameters are quickly accessible. On a similar note, what is the RH in Fig. 2F (current Fig. 3b)? Please, include it within the panel.

We thank the referee for their practical suggestion to render Fig. 2 and current Fig. 3 more self-explanatory. The figures and the captions have been edited accordingly to reflect these suggestions (see previous comment, also reply to comments from Reviewer #1). We hope that these modifications have rendered the graphs more self-explanatory.

2.2. The authors present the permeability of the membranes, an intrinsic property of the material. Based on the methods, the latter is calculated using the standardized protocol put forward in ASTM M95. In my version of ASTM M95, it is stated on page 5 that: "the calculation of permeability is optional and can be done only when the test specimen is homogeneous (not laminated) and not less than 1/2 in. [12.5 mm] thick ...". It seems the SBS/CNC membranes fail on both counts. For example, the thickest of the membranes used in the paper is 100 times thinner than the minimum thickness recommended in the international standard. Can the authors give a justification?

As the reviewer noted correctly, the ASTM **E96** standard states that the calculation of permeability is optional without specifying the reasons why a laminate or a thinner sample than 12.5 mm would not be suitable for such a calculation. For dense membranes (e.g., sheets for packaging materials, gas separation and pervaporation membranes) the permeability is the preferred figure of merit to represent mass transport, if the membrane thickness can be determined accurately without affecting the structure of the membrane and given that the thickness is homogeneous (see for example Refs 41 and 44-47 cited in the manuscript and also R.W. Baker, *et al.*, *J. Membr. Sci.* 348 (2010) 346–352 for dense pervaporation membranes), since it corresponds to the intrinsic properties of the material as also highlighted by the referee. On this basis, we decided to report the permeability instead

of non-intrinsic properties, such as permeance (thickness-dependent) or vapor transmission rates (geometry- and chemical gradient- dependent) using SI Equation 3. To be more precise, we have added an extra note on this to the SI methods (SI Lines 57-60).

2.3. In Fig. 2 c (current Fig. 2e) and f (current Fig. 3b), does “a.u.” mean “arbitrary unit”? The permeability asymmetry is a non-dimensional ratio. It has no unit.

We thank the referee for noting this error, which we have corrected. In fact, the same comment must be made for the RCC and $\tan\delta$. All the pertinent graphs (main Figs. 1-5, Fig. S5, Fig. S6, Fig. S9, Fig. S12) were corrected accordingly.

2.4. The authors never clearly explain why permeability increases with thickness for the SBS/CNC 15% membranes. They simply say “but a dependence on the thickness has been reported for hydrophilic membranes and this was explained with swelling effects”. However, to my knowledge, this is valid for materials with high sorption capacity. Their SBS/CNC membranes at best can absorb 4% water (Fig. 2d) (current Fig. 2f) which would suggest that swelling is minimal.

To explain this behavior, two aspects must be considered. Firstly, a thickness-dependent permeability has previously been reported for homogeneous dense membranes, especially hydrophilic, but also hydrophobic rubbers; this effect has been attributed to thickness-dependent swelling characteristics (see main text Lines 112-115 and Refs 45-48) and could explain the slight increase of the water permeability through neat SBS membranes that is observed when the thickness is increased from 33 to 120 μm . Secondly, in order to cast thicker SBS/CNC membranes using the same concentration of SBS and CNCs in THF, larger volumes of THF are required, which in turn leads to a longer casting-evaporation process and a higher (local) concentration of the CNCs towards the bottom side of the composites. This effect is clearly evidenced by ATR-IR spectra of the bottom side of the SBS/CNC 15 wt% membranes, which reflect that the relative CNC/SBS ratio at the bottom side of the membranes increases with increasing membrane thickness (Fig. S6e-d). Consequently, the available “sorption sites” for water become more prominent towards the bottom side, and this increases the permeability as a function of thickness. To explain this trend better, we have modified the main text (Lines 86-87, 112-115 and 123-124), added cross-references to the supporting figures (Lines 123-124), and updated the figure legend accordingly (SI Lines 349-351).

3. Regarding the data reported in Fig. 3 (current Fig. 4), I appreciate greatly the additional work performed by the authors to establish more firmly the transport asymmetry in intact cuticles. I believe the way they chose the present the new data, i.e. as suppl. mat., works well. Note, however, that in Fig. S12 a, the upper labels should read “Outward tranSport” and “Inward tranSport”.

We thank the referee for their positive comment regarding our additional work on cuticular transport asymmetry and the way these data are presented. The typos in the labels of Fig. S12 have been corrected.

4. I am completely baffled by the mechanical analysis of Fig. 4 (current Fig. 5) and the authors’ attempt to extract from it an explanation for the water transport asymmetry in leaf cuticles. I feel very uncomfortable with the inferences that are made based on these results. First, as the authors noted themselves, the cuticle is a complex material made of at least three distinct compounds (waxes, cutin, polysaccharides) and its structure is both heterogenous and anisotropic. Given this complexity, the interpretation of water transport experiments and mechanical experiments must be done very carefully. Normally, this is done by isolating every single component to understand its behavior. While the authors have made

some attempts to “simplify” their cuticles by removing waxes, they are still dealing with a complex material. Therefore, the suggestion that plasticization of cutin explains the transport asymmetry is, in my view, only one of many other possible explanations. It seems the idea is put forward only because it allows the authors to reconcile their results for the SBS/CNC membranes with those for the cuticle. The results presented in fig. 4 (current Fig. 5) certainly do not provide solid evidence that changes in the cutin layer are the only realistic mechanism for the water transport asymmetry.

The referee’s critique that the results of the humidity-dependent mechanical experiments have been over-interpreted is well taken. We note, however, that this was not at all done to reconcile the results of the SBS/CNC membranes. In fact, we stated clearly that the two membrane types have ultimately different operating mechanisms (main text Lines 203-212).

Since the cross-linked nature of the cutin prevents further “disassembly” of the natural cuticles into the individual components (beyond the demonstrated extraction of the waxes) we opted to reword the conclusions drawn from the moisture-dependent DMA experiments. We wish to point out that such moisture-dependent DMA studies of leaf cuticles are unprecedented and have allowed important insights on the plasticizing effect of water on cuticles.

The plasticizing effect of water on the cuticles as a *whole* is unequivocally reflected by the reduced storage modulus E' upon hydration of the samples, both with and without waxes (Fig. 5c), although—as the reviewer correctly remarks—the result does not allow one to draw a clear conclusion to what extent the various components (polysaccharides, cutin, and other non-extractable components) contribute to membrane stiffness. We have edited the manuscript to reflect this (see main text Lines 256-262). We further note that the “wet” modulus of the wax-free membranes ($E' = 21 \pm 4$ MPa) is comparable to the “dry modulus” of the same membranes at elevated temperatures (e.g., $E' = 32 \pm 13$ MPa, 130 °C). Under both conditions, the DMA reflects the existence of a rubbery plateau that is characteristic of a cross-linked polymer (which again may or may not be cutin) and the data show that the hydration of the cuticles induces similar structural changes as raising the temperature well above the glass transition (see main text Lines 244-250). These findings agree well with previous works using other analytical methods that suggest a plasticizing effect of water on cuticles (main text Refs 53, 56-59) inducing a reduction of the glass transition and phase transformation of cutin (main text Ref 57).

Second, the transport asymmetry measured by the authors is small (not even a factor of 2). Therefore, minor changes in any of the components of the cuticle could account for the observed asymmetry. It is not clear to me that the change must be of a magnitude that can be picked up by a mechanical test and the structural change certainly does not need to be associated with a phase transition.

The qualification of an effect as “small” (or “large”) is a matter of context. We note that the PAF values reported here for olive and ivy cuticles ($PAF^1 = 1.6 - 2.7$, $RH_R = 2\%$, values are inversed to allow for a direct comparison with literature data) are comparable to the PAF values of Nylon:ethyl cellulose bilayer membranes reported by Rogers and co-workers ($PAF = 1.5 - 3.3$, main text Ref 9); in the latter, the asymmetry is caused by swelling and plasticization of the Nylon. Indeed, all other compositionally graded or layered dense membranes reported in literature exhibit asymmetry factors in the range of 1.2 and 4.5. Several authors pointed out that asymmetry factors in heterogeneous dense membranes are typically below 1.5, and that a strong deviation from ideality (i.e., substantial swelling and plasticization) would be required to induce larger asymmetries (A. Peterlin, *et al.*, *Journal of Macromolecular Science, Part B* 6 (1972) 571-582; Petropoulos main text Ref. 11). Thus, the magnitude of the observed asymmetry is certainly as expected and consistent with the

conclusion that the outer cuticular side (but as the reviewer pointed out correctly, *not necessarily* a phase transition of the cutin) displays a considerable dependence on humidity.

Taken together, our permeation data (main text Fig. 4) and the DMA data (main text Fig. 5) *support the conclusion* that the outer cuticular side (but not necessarily the cutin alone) governs the observed asymmetry and that plasticization through sorption of water (from the outer side) is at play, leading to increased molecular mobility and effectively a humidity-responsive permeability, as also noted by others (main text Refs 49, 50, 54, 55). We have reworded the manuscript before (Lines 214-215) and after the DMA discussion (Lines 256-262) accordingly, and also made related changes to the conclusions (Lines 279-281). We hope that our more carefully worded conclusions will be acceptable to the referee.

Reviewer #3

The paper entitled “Asymmetric water transport in olive cuticles and cuticle-inspired, compositionally graded membranes” by Kamtsikakis et al. is an interesting and timely work about the water transpiration mechanisms through plant cuticles, in particular those from olive and ivy leaves. The authors claim that the mechanism depends on the humidity (the hydration of cutin polymer) and support their results by using a model “inspired” by plant cuticles. Such a model can find applications in different industrial uses (i.e. smart packaging). The text is academic and easy to follow and figures are intuitive and very nice (especially 3d models). The experimental section is well described, providing details enough to be reproduced.

We thank the referee for their positive evaluation of our study, the concise summary, and their recommendations, which have helped us to improve this manuscript.

I have some concerns and recommendations that should be addressed before publication:

1.- In my opinion, “ivy cuticles” should be also included in the title.

Mentioning both plant species makes the title, in our opinion, a bit awkward. We have edited it to say “dense leaf cuticles”, which shifts the attention away from the olives. We hope this is acceptable.

2.- I miss a good comparative (maybe a table or something similar) between authors’ model and other cuticle-mimetic systems, indicating their different chemical compositions.

We have in fact cited all relevant and notable cuticle mimicking membranes in the introduction (main text Lines 35-36, Refs 31-35). They are all multilayered systems, and the directionality of their water transport characteristics has not been investigated. To our best knowledge, no water-transport measurements for other graded, dense cuticle-mimicking membranes have been reported. Thus, we see no reason to expand our discussion on previous works (no change to manuscript). However, we have expanded the discussion on our cuticle-inspired system and in relation to cuticles and mass transport (see comments 3 and 4, Lines 50-60).

3.- The participation of waxes, that are essential in the water transport of plant cuticles, is absent in authors’ model. Practically a three-layer system (polysaccharides/cutin/epicuticular waxes) is compared to a two-layer membrane (CNC/SBS). Please, devote some lines to this during discussion.

We thank the referee for their comment, which reflects that our description was indeed unclear. In fact, we did not omit the waxes, but rather the polar functional groups of cutin. We have edited the main text (Lines 50-53) to bring this out better and explain that we “approximated the complex architecture of the cuticle by a compositionally graded two-component nanocomposite. A hydrophobic polymer matrix was used in lieu of the non-polar waxes and lipophilic portions of cutin, hydrophilic cellulose nanoparticles assume the function of the cuticular polysaccharides, and the polar portions of the cutin matrix were omitted (Fig. 1c)”.

4.- Other important aspect for water permeability in polymers is crystallinity. The authors explain well this characteristic for CNC particles, but what about SBS? Is also this polymer crystalline or not? Please, consider that cutin is an amorphous biopolymer, while in the case of waxes the situation is more complex, although they tend to be found as crystals.

SBS is an *amorphous* thermoplastic elastomer composed of poly(styrene) and poly(butadiene) segments (new main text Ref 36), which microphase separate into hard and

relatively impermeable (poly(styrene)) and soft and relatively permeable (poly(butadiene)) domains. Neither of these domain types shows any crystallinity, which is evident from the WAXS data (Fig. S2e), as well as the absence of birefringent domains in the polarized optical microscopy images (Fig. S3a). To bring this out better, we have modified the main text accordingly, and better explain (a) the selection of SBS, (b) the nature of the SBS' domains and how these domains contribute to the permeability (main text Lines 55-60, new Ref 36).

5.- In the conclusions, the authors suggest to change CNC by PVA. Although this modification is correctly reasoned, it could be interesting for the industrial application of membrane, but not for cuticle-inspired systems. Maybe, the use of polysaccharide-based fibers is a best choice for the latter case.

We note that the CNCs employed in the present study are in fact polysaccharide-based fibers, but as noted in the manuscript, their considerable crystallinity suppresses transport *through* the nanoparticles, and water transport occurs primarily or even exclusively along their surface (main text Lines 62-65). Thus, polysaccharide fibers having a lower crystallinity might indeed induce a stronger asymmetry, increase the permeability, and simultaneously represent better the polar polysaccharides present in natural cuticles better than synthetic PVA fibers. We have edited the concluding section to reflect this (main text Lines 277-278).

6.- The chemical differences between SBS and cutin are very important. Among them, SBS is a linear polymer (a thermoplastic), while cutin shows a complex behavior typical of thermosets with no meltin point and lack of solubility. Can also these differences influence the water permeation?

The referee raises an important point, which we believe to have addressed in the manuscript by highlighting that the SBS hardly takes up water, exhibits humidity-independent properties and that therefore the transport in the artificial membranes is governed by the polar CNC filler rather than the SBS matrix in the artificial system (see main text Lines 215-223). We have further noted that these differences in the transport behavior are presumably “linked to a crucial role that the cutin matrix plays in the cuticles, and which is absent in the simplified SBS/CNC system” (main text Lines 214-224), and believe no further edits are needed.

Other changes

The abstract was shortened from 210 to 143 words in order to comply with the journal's formatting guidelines.

Sub-headings were added in the results section of the main text to comply with the journal's formatting guidelines.

After adding the schematics of the experimental setups (see above) to Fig. 2, we had to split off original Figs. 2e,f into new Fig. 3 (content of the figure is the same as previous Fig. 2e,f), to prevent Fig. 2 from becoming too large. Thus, original Figs. 3 and 4 were renumbered into Figs. 4 and 5.

We removed the equations of *PAF* from Figs. 2, 3 and 4 to declutter the graphs. We have added the title “olive cuticles” in Fig. 4a-b to homogenize with Fig. 4c. We have changed the colors in Fig. S8d and modified Fig. S12a y-axis to align the graph with panel (c).

We made linguistic changes on Lines 20, 48, 49, 50-53, 77, 127, 131, 207, 214-215, 231-232, 267, 390-391, 397-398, 404-408, and SI 68-69, 88, 394-396 and in the SI Tables S3, S4 and S6 to reflect the suggested changes by the referees.

We have added the word “Supplementary” to the link to the SI figures and tables.

REVIEWERS' COMMENTS

Reviewer #2 (Remarks to the Author):

I thank the authors for their patience and willingness to make the necessary changes to improve the impact of their paper. They have now responded positively to all of my concerns.

Reviewer #3 (Remarks to the Author):

All my comments have been properly addressed.